# Opposite initialization to novel cues in dopamine signaling in ventral and posterior striatum in mice

William Menegas, Benedicte M Babayan, Naoshige Uchida, Mitsuko Watabe-Uchida*

Department of Molecular and Cellular Biology, Center for Brain Science, Harvard University, Cambridge, United States

**Abstract** Dopamine neurons are thought to encode novelty in addition to reward prediction error (the discrepancy between actual and predicted values). In this study, we compared dopamine activity across the striatum using fiber fluorometry in mice. During classical conditioning, we observed opposite dynamics in dopamine axon signals in the ventral striatum ('VS dopamine') and the posterior tail of the striatum ('TS dopamine'). TS dopamine showed strong excitation to novel cues, whereas VS dopamine showed no responses to novel cues until they had been paired with a reward. TS dopamine cue responses decreased over time, depending on what the cue predicted. Additionally, TS dopamine showed excitation to several types of stimuli including rewarding, aversive, and neutral stimuli whereas VS dopamine showed excitation only to reward or reward-predicting cues. Together, these results demonstrate that dopamine novelty signals are localized in TS along with general salience signals, while VS dopamine reliably encodes reward prediction error.

## Introduction

Animals respond to new stimuli in a characteristic way across species, historically characterized as an 'orienting reflex' or a 'what is it reflex' (*Pavlov and Anrep, 1927*; *Sechenov, 1935*; *Sokolov, 1963*). Detection of novel stimuli is advantageous for survival because novel stimuli can signal potential rewards or potential threats. Orienting towards a novel stimulus and understanding it through exploration can allow future exploitation of potential rewards. In addition to behavioral advantages, novelty detection is fundamental for computation in our brain. For example, novelty detectors, or 'novelty filters' (*Kohonen and Oja, 1976*; *Marsland et al., 2002*), can reduce the amount of total information so that we can focus on unexpected perceptions as inputs to pay attention to and to learn from. Indeed, behavioral studies have repeatedly shown that both humans and other animals have enhanced memory for novel items (*Kishiyama et al., 2009*; *Restorff, 1933*).

Physiologically, it is widely accepted that novelty responses are distributed over a network of many brain areas (*Courchesne et al., 1975*; *Kishiyama et al., 2009*; *Knight, 1996*). Among these, single unit recordings have shown that dopamine neurons in the midbrain increase their firing in response to the presentation of a novel stimulus in several species and behavioral paradigms (*Horvitz et al., 1997*; *Ljungberg et al., 1992*; *Schultz, 2015*; *Steinfels et al., 1983*).

As animals experience the repeated association of a stimulus and a reward, they learn to expect the reward when the stimulus is presented (*Pavlov and Anrep, 1927*). Dopamine neurons are thought to be the neural substrate underlying this type of learning because they signal reward prediction error: the difference between actual and expected reward values (*Bayer and Glimcher, 2005*; *Bromberg-Martin et al., 2010*; *Clark et al., 2012*; *Cohen et al., 2012*; *Schultz et al., 1997*). These neurons are thought to guide decision-making by broadcasting this information to many

*For correspondence: mitsuko@ mcb.harvard.edu

**eLife digest** New experiences trigger a variety of responses in animals. We are surprised by, move towards, and often explore new objects. But how does the brain control these responses?

Dopamine is a molecule that controls many processes in the brain and plays critical roles in various mental disorders, diseases that affect movement, and addiction. Rewarding experiences (like a glass of cold water on a hot day) can trigger dopamine neurons and studies have also shown that dopamine neurons respond to new experiences. This suggested that novelty may be rewarding in itself, or that novelty may signal the potential for future reward. On the other hand, it may be that different groups of dopamine neurons play different roles in responding to new or rewarding experiences.

In 2015, it was reported that dopamine neurons connected to the rear part of an area in the brain called the striatum receive signals from different parts of the brain than most other dopamine neurons. The dopamine neurons connected to this "tail" of the striatum preferentially received inputs from regions involved in arousal rather than reward, suggesting that they may have a unique role and transmit a different type of information.

Now, Menegas et al. have shown that dopamine signals in different areas of the striatum separate reward from novelty and other signals in mice. The results demonstrate that new odors activate dopamine neurons projecting to the tail of the striatum, but that this activity fades as the novelty wears off (as the mice learn to associate the odor with a particular outcome). By contrast, dopamine neurons projecting to the front of the striatum do not respond to novelty, but rather become more active as mice learn which odors accompany rewards (only responding to odors that predict reward). The experiments also show that dopamine neurons in the tail of the striatum encode information about the importance of a stimulus.

Together, these findings indicate that some of the roles dopamine plays in the brain may not be related to reward, but are instead linked to the novelty and importance of the stimulus. The next challenge will be to find out how the separate reward and novelty signals in dopamine neurons relate to the animals' behavior. This may help us to better understand dopamine-related psychiatric conditions, such as depression and addiction.

regions of the forebrain and reinforcing behaviors that lead to reward (*Barto et al., 1999*; *Dayan and Niv, 2008*; *Haber, 2014*; *Montague et al., 2004*; *Steinberg et al., 2013*).

Novelty responses in dopamine neurons (*Horvitz et al., 1997*; *Schultz, 2015*) were initially puzzling because animals cannot know whether a novel stimulus will reliably predict an outcome with a positive or negative value. One hypothesis was that dopamine neurons take an optimistic approach toward novel stimuli, assuming that they will predict a valuable outcome until proven wrong (*Hazy et al., 2010*; *Kakade and Dayan, 2002*). This 'optimistic initialization' in dopamine neurons may have advantages. For example, the novelty responses in dopamine neurons may induce orienting behaviors towards novel stimuli, similar to dopamine responses to reward or reward-predicting cues that induce orienting behaviors (*Hazy et al., 2010*; *Kakade and Dayan, 2002*). Further, dopamine novelty responses may allow computational exploration (*Dayan and Sejnowski, 1996*), or storage of the novel stimulus in working memory (*Braver and Cohen, 1999*), so that animals have a better chance to associate novel stimuli to potential rewards (*Hazy et al., 2010*; *Kakade and Dayan, 2002*). However, these hypotheses do not necessarily fit with conflicting observations of animals' behavioral responses to novel stimuli (*Gershman and Niv, 2015*). Indeed, depending on the experimental context, animals sometimes approach and sometimes avoid novel options compared to familiar ones (*Gershman and Niv, 2015*).

One explanation for why some dopamine neurons respond to novel stimuli could be that some subpopulations of dopamine neurons are not strictly related to reward prediction error coding. Recent studies have shown that there is substantial diversity among dopamine neurons at the molecular level (*Grimm et al., 2004*; *Lacey et al., 1989*; *Lammel et al., 2008*; *Roeper, 2013*) as well as in their activity (*Brischoux et al., 2009*; *Bromberg-Martin et al., 2010*). For example, single unit recordings in monkeys showed that some dopamine neurons are inhibited by aversive outcomes and

others are excited by them (*Matsumoto and Hikosaka, 2009*). This suggests that there are distinct types of dopamine neurons and that some do not encode pure value. Instead, the data suggest that some dopamine neurons encode value and others might encode 'motivational salience' (the absolute value of 'value').

Recent anatomical studies have revealed that dopamine neurons with different projection targets are embedded in separate circuits. The entire set of inputs to dopaminergic nuclei includes a large number of regions (*Geisler et al., 2007*; *Geisler and Zahm, 2005*). Neural circuit tracing using a modified rabies virus (*Wickersham et al., 2007*) enabled us to specifically label the monosynaptic inputs onto dopamine neurons, revealing that the ventral tegmental area (VTA) and the substantia nigra compacta (SNc) dopamine neurons receive slightly different inputs (*Watabe-Uchida et al., 2012*). More recent studies have shown that dopamine neurons with different projection targets receive different inputs (*Beier et al., 2015*; *Lerner et al., 2015*; *Menegas et al., 2015*). Specifically, we found that dopamine neurons projecting to the posterior 'tail' of the striatum (TS) have unique inputs compared to dopamine neurons projecting to many other brain regions, including the ventral striatum (VS), suggesting that these neurons could have a distinct function (*Menegas et al., 2015*).

Based on our previous anatomical findings, in this study, we compared the dopamine axon activity in VS and TS while mice learned new odor-outcome associations (we will call the bulk calcium signal that we observed from the axons of DAT+ midbrain dopamine neurons in the striatum 'VS dopamine' and 'TS dopamine' in the following sections). Our results revealed opposite dynamics for learning new cue-outcome associations in VS dopamine and TS dopamine. We observed a large response to novel cues in TS dopamine which subsequently decreased over the course of associative learning. On the other hand, we saw no response to novel cues in VS dopamine. Instead, VS dopamine gradually developed responses to reward-predicting cues during learning. These findings revealed that dopamine novelty coding is localized to the posterior part of the striatum, while VS dopamine faithfully encodes reward prediction error. Thus, novelty responses in dopamine may be better formalized separately from the reward prediction error (RPE) framework, rather than being included in the RPE framework.

## Results

### Recording activity from dopamine axons in the striatum

We used optical fiber fluorometry (fiber photometry) (*Kudo et al., 1992*) to record bulk calcium signals from the axons of midbrain dopamine neurons projecting to several regions of the striatum (*Kim et al., 2016*; *Parker et al., 2016*). We chose four regions: the ventral striatum (VS), dorsomedial striatum (DMS), dorsolateral striatum (DLS), and the posterior tail of the striatum (TS) (*Figure 1*). To measure activity from dopamine axons in these regions, we infected midbrain dopamine neurons with a genetically encoded calcium indicator, GCaMP6m (*Akerboom et al., 2012*; *Chen et al., 2013*). To target dopamine neurons specifically, we injected a cre-dependent virus (AAV-flex-GCaMP6m) into both the VTA and SNc of transgenic mice expressing Cre recombinase under the control of dopamine transporter (DAT-cre mice) (*Bäckman et al., 2006*) crossed with reporter mice expressing red fluorescent protein (tdTomato) (Jackson Lab).

We chronically implanted optic fibers into the striatum of these mice to deliver 473 nm and 561 nm light and collect GCaMP and tdTomato signals (*Chen et al., 2012*; *Gunaydin et al., 2014*; *Kim et al., 2016*) (*Figure 1A*, *Figure 1—figure supplement 1*). For these experiments, we continuously excited GCaMP with 473 nm light and continuously recorded GCaMP emission (*Figure 1—figure supplement 2*). We recorded from 65 fibers total, targeted to either VS (n = 25), DMS (n = 8), DLS (n = 8), or TS (n = 24) (*Figure 1B*, *Figure 1—figure supplement 3*). In fixed tissue, we observed fluorescence from GCaMP6m+ axons primarily in the striatum of these mice (*Figure 1—figure supplement 4*).

Mice were presented with odors paired with water delivery or no outcome (*Figure 1C*). In some experiments, odors were paired with an aversive air puff or a mild tone. Infrequently, mice also received unpredicted water, air puff or tone (≤10% of trials). After training, mice licked with an increased frequency in response to the reward-predicting odor (anticipating the reward), but not in response to odors that predicted no outcome (*Figure 1D*), indicating that mice had learned an association between an odor and reward. We observed large responses to unpredicted reward in

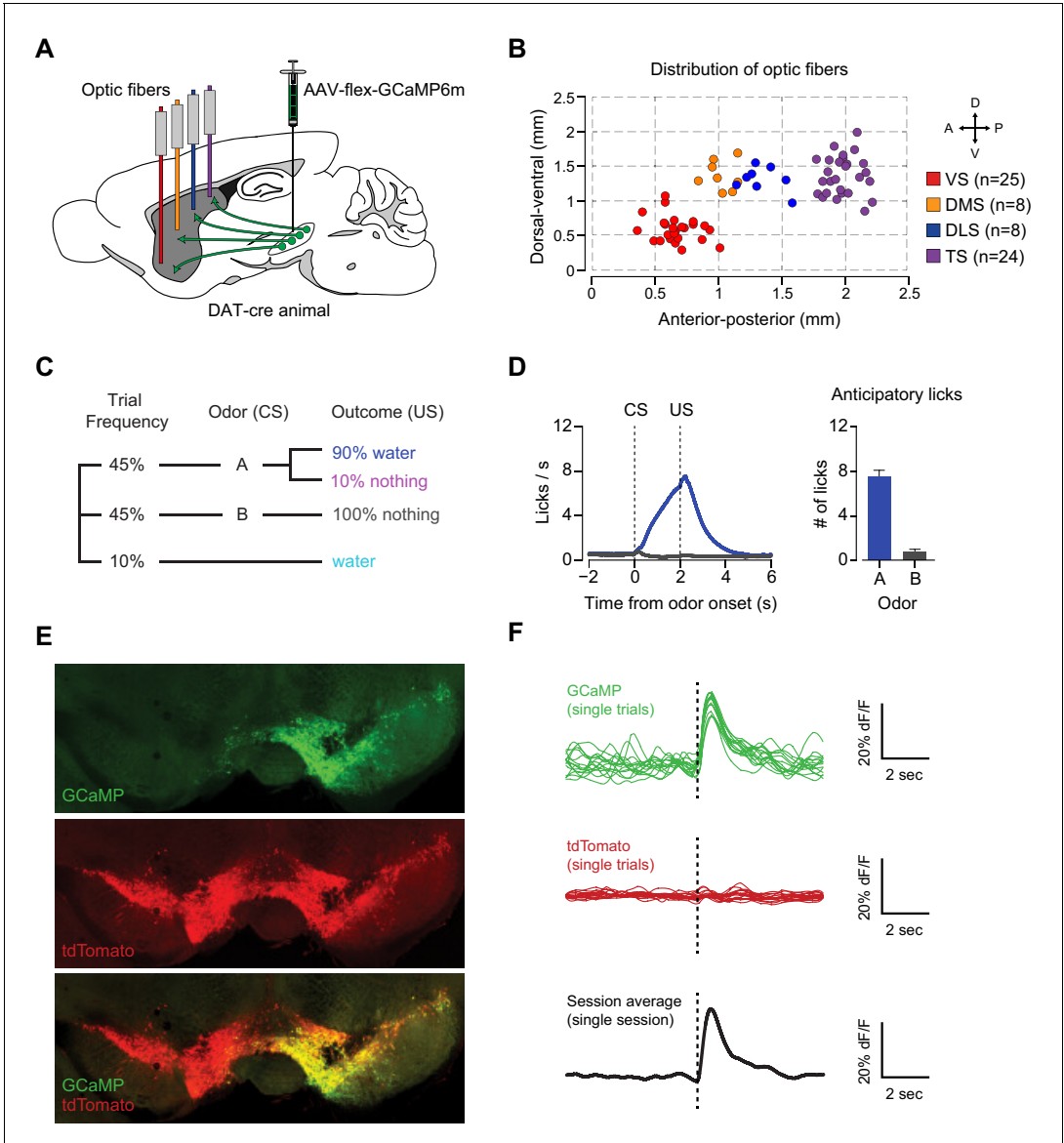

**Figure 1.** Recording dopamine activity across the striatum using fiber fluorometry. (A) Schematic of GCaMP virus injection and optic fiber implantation sites. Detailed schematic of recording setup is shown in *Figure 1—figure supplement 1*. Sample raw data are shown in *Figure 1—figure supplement 2*. (B) Distribution of optic fibers (sagittal max-projection) used for recording labeled red (VS), orange (DMS), blue (DLS), and purple (TS) with dotted lines denoting ½ mm increments. Coronal sections are shown in *Figure 1—figure supplement 3*. (C) Schematic of the basic trial structure. An odor cue (CS) (1 s duration) is followed by an outcome (US) or no outcome after 1 s delay, followed by a random inter-trial interval (ITI) of 6–12 s. At a low frequency, unexpected outcomes are also delivered. (D) Licking in response to odors predicting reward (blue) or nothing (grey). Odor onset is t = 0 and water delivery time is t = 2, so anticipatory licking occurs between t = 0 and t = 2 (quantified on the right). (E) An example of GCaMP virus infection. Green indicates AAV-flex-GCaMP6m infection (top), red indicates genetically encoded tdTomato in DAT-cre-expressing neurons (middle), and the bottom panel is an overlay of the two signals. Labeled axons in the striatum are shown in *Figure 1—figure supplement 4*. (F) Example single trial responses to unpredicted water from GCaMP (top) and tdTomato (middle) from a single session in a mouse with a fiber implanted in VS. The average GCaMP signal across trials in that session are plotted in the bottom panel.

The following figure supplements are available for figure 1:

**Figure supplement 1.** GCaMP6m recording and example traces.

**Figure supplement 2.** Example recording sessions.

**Figure supplement 3.** Distribution of recording fibers.

*Figure 1 continued on next page*

*Figure 1 continued*

**Figure supplement 4.** Midbrain dopamine axon distribution in the striatum.

GCaMP, but not tdTomato, signals (example traces shown in *Figure 1E–F* and *Figure 1—figure supplement 1B*) and recorded from the same fibers over the course of several weeks with relative stability (*Figure 1—figure supplement 1C*).

We identified the fiber implant sites by clearing brains using CLARITY (*Chung and Deisseroth, 2013*), imaging them as intact volumes using a light sheet microscope (*Tomer et al., 2014*), and aligning them to a single reference space (*Menegas et al., 2015*). We categorized the location of fibers in the dorsal striatum into the DMS, DLS, or TS based on their medial-lateral and anterior-posterior positions (see Materials and methods). VS fibers were spread throughout the core and lateral shell of the ventral striatum (*Figure 1—figure supplement 3*). TS fibers were located near the posterior end of the dorsal striatum (*Figure 1—figure supplement 3*). We will focus on VS and TS dopamine, because VS and TS dopamine displayed the most contrasting input patterns in our previous anatomical study (*Menegas et al., 2015*).

## Excitation to novel cues in TS dopamine

In order to examine dopamine activity in VS and TS during associative learning, we recorded both during the initial learning of new odor-outcome associations (first time association, *Figure 2* and *Figure 3*) and also during repeated learning where animals experienced new associations every day (*Figure 4*).

For initial training, mice were first habituated in a recording set-up with head-fixed preparation for 2–3 days (see Materials and methods). After this initial habituation, animals were presented with four randomly interleaved trial types: (1) unpredicted water, (2) odor predicting water followed by water delivery, (3) odor predicting water followed by no outcome (omission), or (4) odor predicting no outcome.

We first compared VS dopamine and TS dopamine responses to an odor predicting no outcome over the course of the session (*Figure 2*). At the time of novel odor presentation, TS dopamine was excited very strongly by new odors (*Figure 2D*). By contrast, VS dopamine did not respond to novel odors – not even on the very first trial (*Figure 2A*). We examined 13 animals with fibers implanted into VS and 12 animals with fibers implanted into TS. TS dopamine showed significant excitation above baseline following the presentation of novel odors ($p = 1.76 \times 10^{-6}$, t-test, n = 12 animals, *Figure 2G*), whereas responses to novel odors in VS dopamine were not significantly different from baseline ($p = 0.354$, t-test, n = 13 animals, *Figure 2G*).

The responses to novel odors in TS dopamine decreased significantly over the course of the first 30 odor presentations ($p = 5.10 \times 10^{-11}$, repeated measures ANOVA, n = 12 animals, *Figure 2E–F*). The responses after five odor representations (6–10 trials) were significantly smaller than the responses in the first five trials ($p = 1.45 \times 10^{-4}$, paired t-test, n = 12 animals). To determine whether this decrease in signal was caused by a decrease in effective odor concentration within sessions, we compared responses to fast-decaying and slow-decaying odors (*Figure 2—figure supplement 1*). TS dopamine responses to novel odors, both fast-decaying and slow-decaying odors, decreased over the course of a session (*Figure 2—figure supplement 1F*), whereas TS dopamine responses to familiar odors did not change (*Figure 2—figure supplement 2C*). In VS, no responses were seen over the course of the session (*Figure 2B–C*). Both VS dopamine and TS dopamine showed excitation to water (responses to the first presentation of unpredicted water are shown in *Figure 2G*).

To determine whether the signals we observed could have been caused by a movement-related artifact, we used video analysis to quantify the total body movements of mice performing the task (*Figure 2—figure supplement 3*). We found that mice did not show major body movements in response to odors predicting no outcome (*Figure 2—figure supplement 3B*) or novel odors (*Figure 2—figure supplement 3C*), although they performed a stereotypical approach behavior in response to odors predicting reward (*Figure 2—figure supplement 3A*). To determine whether changes in signal intensity observed within a session were likely to have been related to bleaching,

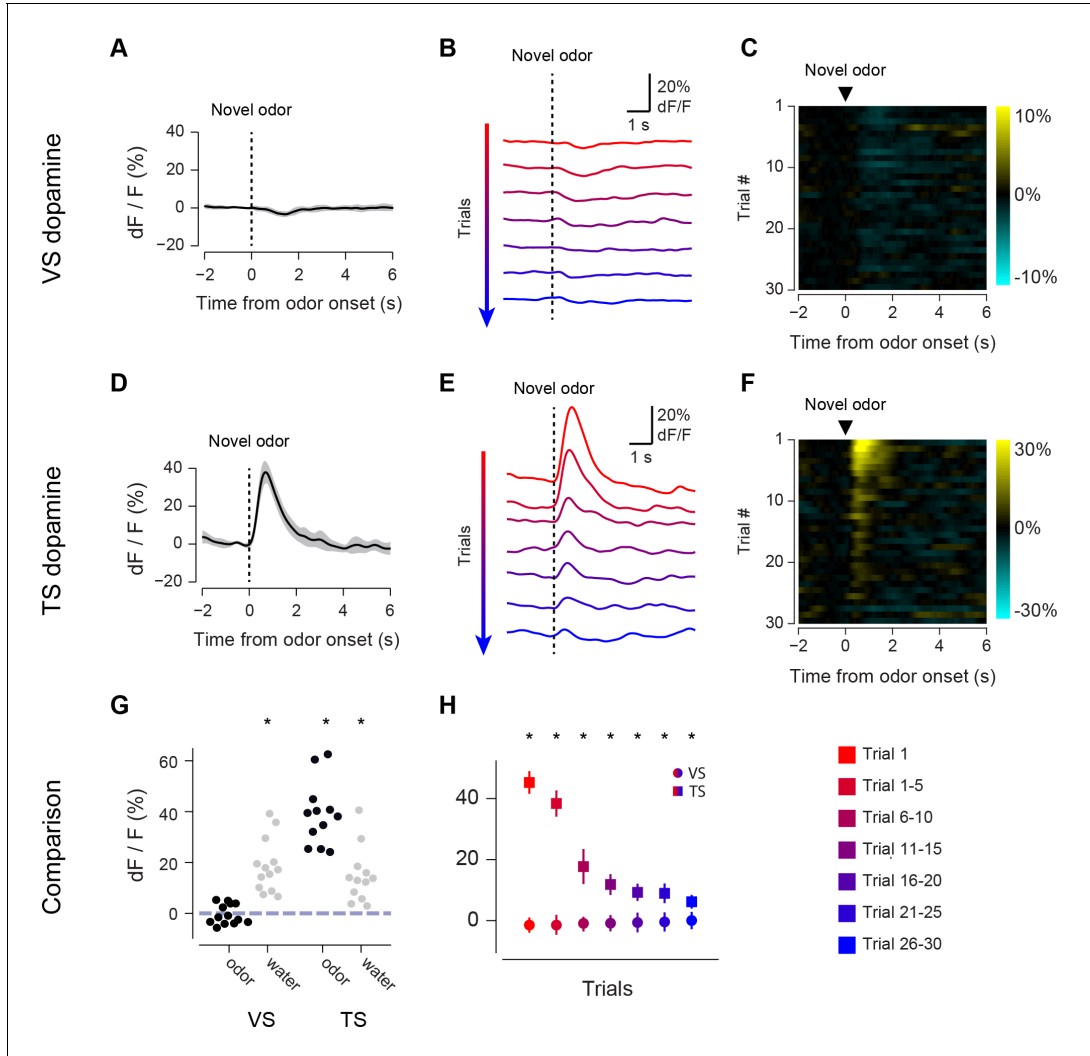

**Figure 2.** Responses to novel odors in VS and TS dopamine. Comparison of VS and TS responses to novel odors in naïve animals. (**A**) Average response to the first presentation of a novel odor in VS dopamine, with SE bars. (**B**) Average responses over the course of the first 30 trials are shown in bins of 5 trials. (**C**) A heat map of responses to a novel odor over the course of a single session (each row is one trial) with yellow indicating an increase in signal and cyan indicating a decrease in signal. (**D–F**) TS dopamine responses to novel odors, plotted as in **A–C**. (**G**) Comparison of first-trial water responses in VS and TS (left) and first-trial responses to novel odors (right). See Materials and methods. (**H**) Time course of responses to a novel odor in VS (circles) and TS (squares) over the course of 30 trials in bins of 5 trials. This data was analyzed based on odor decay rates to show that there was no large effect of odor decay in *Figure 2—figure supplement 1*. Motion artifacts were examined in *Figure 2—figure supplement 3*. GCaMP signal decay was measured in *Figure 2—figure supplement 2*. Finally, response latencies are shown in *Figure 2—figure supplement 4*.

The following figure supplements are available for figure 2:

**Figure supplement 1.** PID measurements of odor decay rates.

**Figure supplement 2.** GCaMP response decay within sessions.

**Figure supplement 3.** Animal body movement during trials.

**Figure supplement 4.** Latency of GCaMP responses to novel odors in TS.

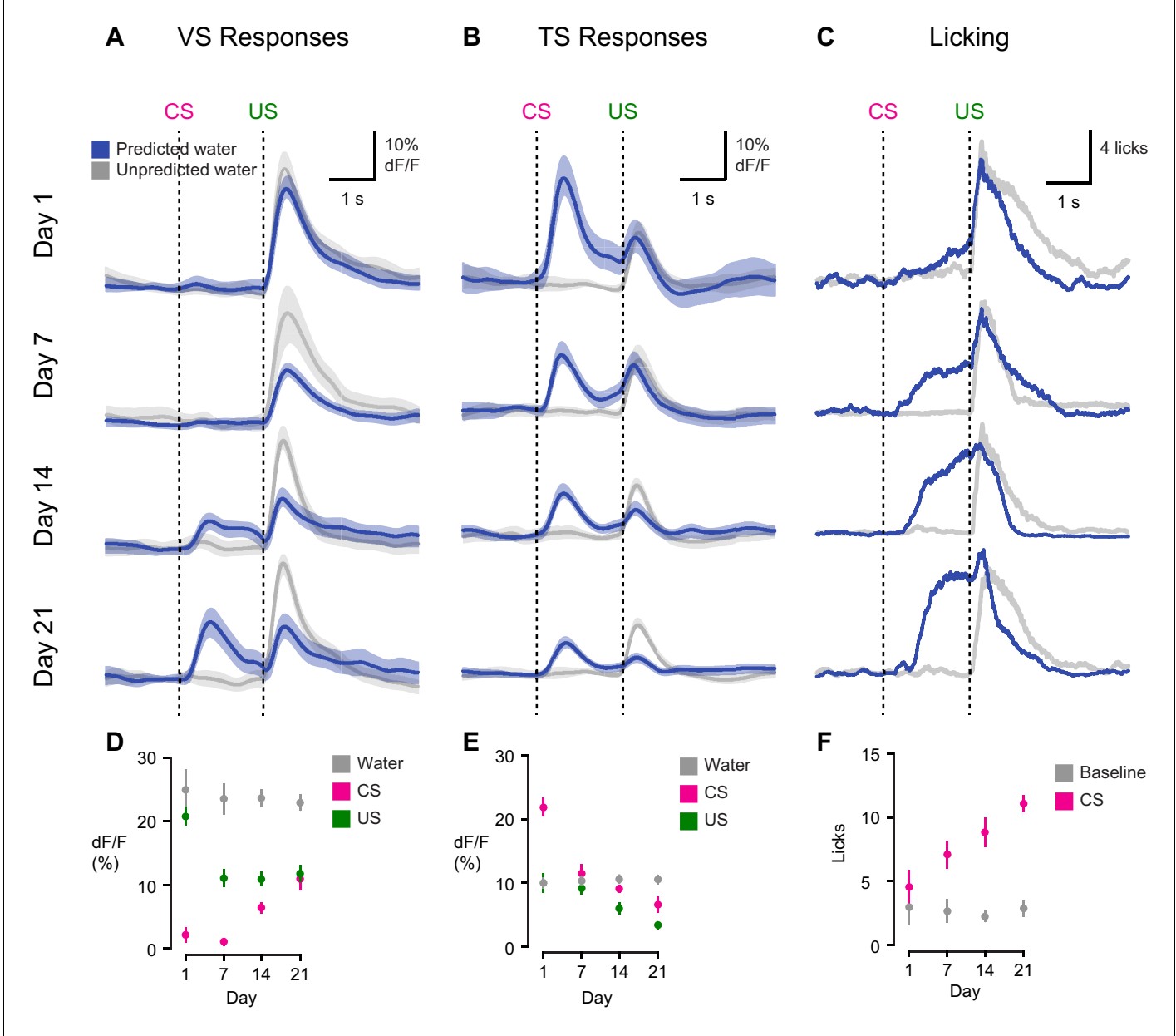

**Figure 3.** Opposite dynamics of VS and TS dopamine during initial learning of new odor-reward associations. Learning dynamics for VS dopamine (**A**) and TS dopamine (**B**) over the course of 3 weeks of training, as naïve animals learn an association between an odor and reward. Odor onset (CS) and water delivery time (US) are shown as dotted lines. Responses are compared on day 1, day 7, day 14, and day 21. The average traces are plotted in blue (predicted reward) and black (unpredicted reward), with the standard error of the mean (SEM). Individual animals' responses can be found in *Figure 3— figure supplement 1*. (**C**) Average licking in response to reward-predicting odor (blue) compared to average licking in response to unexpected reward (black). (**D**) A quantification of the CS and US responses in VS from the above traces, over training compared to responses to unexpected water (black). (**E**) A quantification of the CS and US responses in TS from the above traces, over training, compared to responses to unexpected water (black). (**F**) The average number of anticipatory licks in the period between odor presentation and water delivery, compared over days of training.

The following figure supplement is available for figure 3:

**Figure supplement 1.** Individual traces during initial learning of new odor-reward associations.

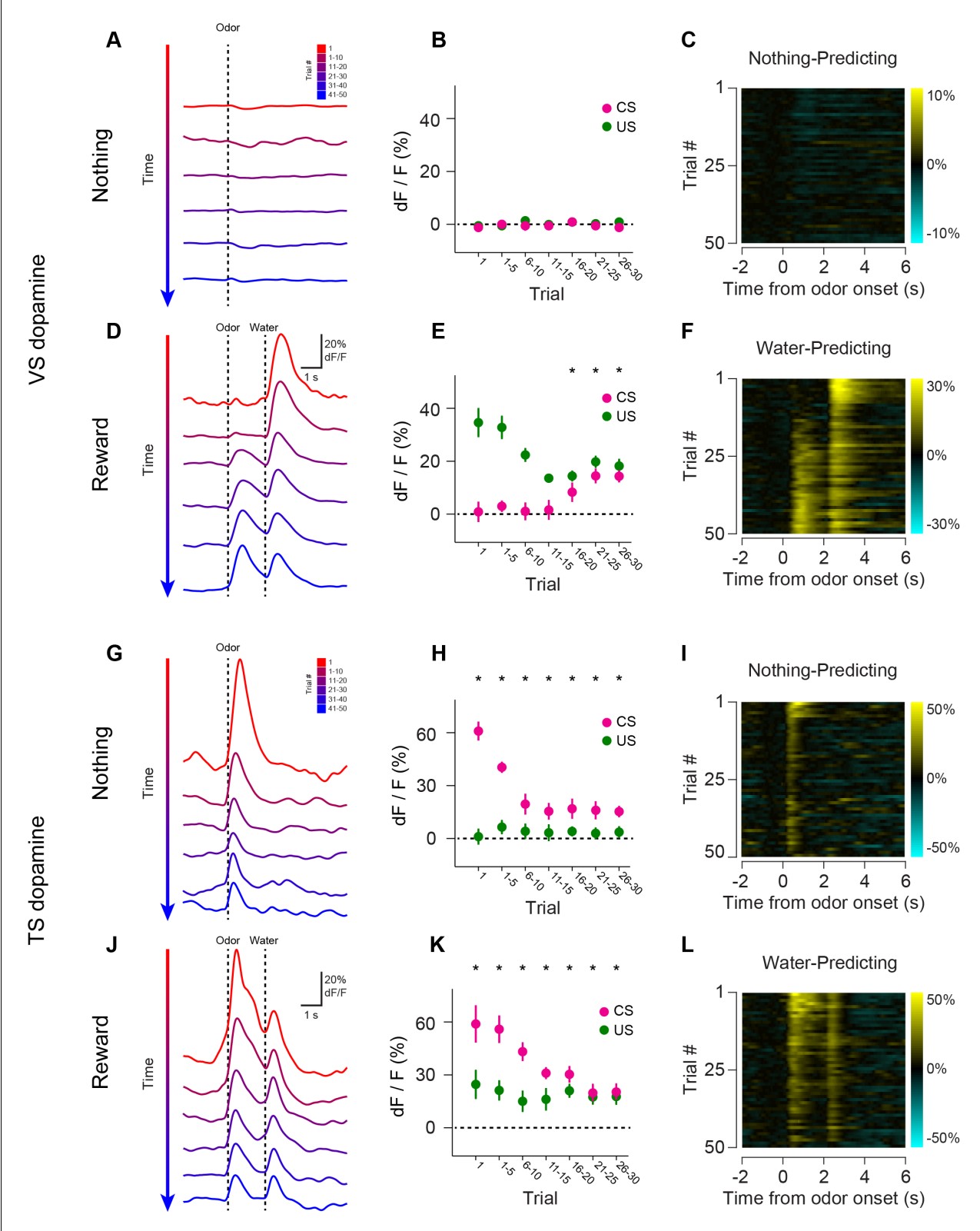

**Figure 4.** Opposite dynamics of VS and TS dopamine during repeated learning of new odor-reward associations. Responses to new cues predicting nothing (A–C) or water (D–F) in VS. Responses to new cues predicting nothing (G–I) or water (J–L) in TS. Responses to aversive air puffs are found in *Figure 4—figure supplement 1* and quantified in *Figure 4—figure supplement 2*. In the panels on the left, trials are color-coded such that red indicates the first trial and blue indicates the last trials of the session. Trials were quantified in bins of 5 trials. The middle panels show the average CS

*Figure 4 continued on next page*

*Figure 4 continued*

(magenta) and US (green) responses over the course of a session, again quantified in bins of 5 trials. The panels on the right are heat maps, where every line is a single trial. In these heat maps, yellow indicates an increase in signal and cyan indicates a decrease. Odor discrimination latency is quantified in *Figure 4—figure supplement 3*.

The following figure supplements are available for figure 4:

**Figure supplement 1.** Opposite dynamics of VS and TS dopamine during repeated learning of new odor-puff associations.

**Figure supplement 2.** Dynamics of responses to puff-predicting odors.

**Figure supplement 3.** Novel odor discrimination latency in TS dopamine.

we compared responses to free water and familiar odors over the course of sessions (*Figure 2—figure supplement 2*). We found that both VS dopamine responses (*Figure 2—figure supplement 2A*) and TS dopamine responses (*Figure 2—figure supplement 2B*) to free water remained constant within sessions. Similarly, TS dopamine responses to familiar odors predicting no reward remained constant as well (*Figure 2—figure supplement 2C*).

## Opposite initialization to novel cues in VS and TS dopamine

On the first day of odor-outcome association learning, TS dopamine was strongly excited by new odors (*Figure 3*, *Figure 3—figure supplement 1*). The responses to novel odors that predicted water were significantly larger than responses to unpredicted water itself ($p = 1.36 \times 10^{-4}$, paired t-test, n = 19 animals, *Figure 3B*). Dopamine responses to odors gradually decreased over 21 days ($p = 1.095 \times 10^{-8}$, n = 19 animals, repeated measures ANOVA). The responses to water-predicting odors in TS dopamine were significantly smaller on the seventh day of training than on the first day ($p = 1.93 \times 10^{-5}$, n = 19 animals, day 1 vs day 7, *Figure 3B*). On the other hand, responses to predicted water did not change significantly ($p = 0.641$ paired t-test, n = 19 animals, day 1 vs day 7, *Figure 3B*).

By contrast, VS dopamine did not respond to novel odors predicting water (*Figure 3*, *Figure 3—figure supplement 1*). Instead, we observed an initially large excitation in response to water itself (*Figure 3A*). Over the course of the first 7 days, responses to predicted water (US responses) significantly decreased ($p = 1.96 \times 10^{-4}$, paired t-test, n = 10 animals, *Figure 3A*). Notably, this was independent of any CS response developing. VS dopamine did not display significant responses to odor cues that predicted reward on the first day ($p = 0.099$, t-test, n = 10 animals, CS responses compared to baseline on day 1) or on day 7 ($p = 0.054$, t-test, n = 10 animals, CS responses compared to baseline on day 7) (*Figure 3D*). In fact, responses to reward-predicting cues appeared only after 2 weeks of training (*Figure 3A*). Of note, responses to unpredicted water remained constant over the course of learning (*Figure 3A–B*). Anticipatory licking gradually increased in frequency over the course of learning ($p = 0.0052$, repeated measures ANOVA, n = 10 animals, *Figure 3C*).

We tested whether repeated training affected the observed pattern for novel cues and reward signaling in VS and TS dopamine. We trained nine mice with VS fiber implants and 11 mice with TS fiber implants by introducing a new odor paired with a reward or no outcome every day for a week, and then measured dopamine activity while learning new odor-water or odor-nothing associations (*Figure 4*). We found that repeated training with odor-reward associations did not change responses to new odors in VS dopamine or TS dopamine. VS dopamine did not respond to new odors ($p = 0.8749$, t-test, n = 9 animals, trial one or $p = 0.322$, t-test, n = 9 animals, trial 1–5 vs baseline, *Figure 4B–E*) and TS dopamine strongly responded to new odors ($p = 0.0059$, t-test, n = 11 animals, trial one or $p = 0.0027$, t-test, n = 11 animals, trial 1–5 vs baseline, *Figure 4H–K*). Indeed, TS dopamine showed excitation to 91% of new odor presentations (response in trial one vs baseline). Dopamine axon signals in mice repeatedly trained on learning odor-outcome contingencies displayed the same trends in the dynamics, but at a much faster rate: within a single session (*Figure 4*) rather than over the course of weeks (*Figure 3*). VS dopamine showed a decrease in US response followed by an increase in CS response, with no response to novel stimuli (*Figure 4A–F*). TS dopamine decreased responses to either novel odor (nothing-predicting or water-predicting) (*Figure 4G–L*).

To better understand the dopamine response to novel odors, we also paired new odors with an aversive air puff in these well-trained mice (*Figure 4—figure supplement 1*). As in the cases of novel odors predicting water or nothing, VS dopamine showed no odor responses (*Figure 4—figure supplement 1A–C*) and TS dopamine responded strongly to the novel odor (*Figure 4—figure supplement 1D–F*). Notably, the decrease in TS dopamine response to novel odors predicting air puff was much smaller than the decrease of TS dopamine response to novel odors predicting no outcome (*Figure 4—figure supplement 2*), indicating that the dynamics of the response depend on what the novel odor cue predicts.

Anticipatory licking in response to the rewarded odor increased after a few trials (p = 0.0387, paired t-test, n = 20 animals, trial 1–5 water CS lick vs baseline) (*Figure 5*, *Figure 5—figure supplement 1*). The animals showed differences in anticipatory licking frequency depending on cues within 10 trials of training (p = 0.000259, paired t-test, n = 20 animals, trial 6–10 water CS lick vs nothing CS lick), indicating learning of the outcomes of the odor cue (*Figure 5C*). VS dopamine did not show differences in responses to cues before 15 trials (p = 0.7736, paired t-test, n = 9 animals, water CS vs nothing CS trial 11–15, *Figure 5D*), whereas responses to predicted water decreased quickly (p = 0.020, paired t-test, n = 9 animals, trial 1–5 vs 6–10, *Figure 5E*). Plotting the CS and US responses as a function of anticipatory licks (rather than time) showed that mice behaviorally responded to reward-predicting odors faster than VS dopamine CS responses developed, while TS dopamine CS responses were present in all trials (*Figure 5—figure supplement 2*). TS dopamine decreased responses to cues depending on what the cue predicted within five trials (p = 0.0213, paired t-test, n = 11 animals, water CS vs nothing CS trial 1–5, *Figure 5D*) and the difference became smaller later in the session.

We examined the temporal dynamics of the responses to novel odors in TS dopamine. The median onset latency of responses to novel odors in TS dopamine was 140 ms (*Figure 2—figure supplement 4*) and the median onset latency of discrimination between novel odors and familiar odors in TS dopamine was 170 ms (*Figure 4—figure supplement 3*).

In summary, dopamine axon signals in VS and TS showed opposite initialization while learning stimulus-outcome relationships. Dopamine axon signals in TS showed strong excitation to novel cues that gradually decreased, whereas dopamine axon signals in VS did not respond to cues with an unknown outcome and instead gradually developed cue responses to odors reliably predicting reward. Further, TS dopamine quickly discriminated cues, resulting in differential decrease rates of responses to novel cues depending on the predicted outcome (air puff, water, or nothing).

## Responses to rewarding and non-rewarding stimuli in VS and TS dopamine

In order to understand the relationship between novelty responses and value coding, we next examined responses to rewarding and non-rewarding stimuli in VS dopamine and TS dopamine. Mice were trained to associate odors with water or with no outcome. After several weeks of this training, in some sessions, trials with odors predicting either a mild tone (55 dB) or an air puff were interleaved in addition to trials with water and trials with no outcome. We chose a very mild tone with a similar intensity to the background noise in the room to try to minimize the aversiveness of this stimulus. To estimate the aversiveness of auditory stimuli, we measured the behavioral responses to tones of different volumes in a different set of mice (*Figure 6—figure supplement 1*). We found that quiet tones did not cause freezing. When comparing the VS dopamine and TS dopamine responses to all stimuli, we observed that VS dopamine showed excitation only to reward and reward-predicting cues (*Figure 6A–C*, *Figure 6—figure supplement 2*), while TS dopamine was excited in response to a variety of stimuli including water, tone, air puff, odor cues predicting any of these outcomes, and also odor cues predicting no outcome (*Figure 6D–F*, *Figure 6—figure supplement 2*).

We next examined reward prediction error coding, which consists of three key characteristics: (1) larger responses to reward-predicting cues than unrewarded cues, (2) smaller responses to predicted rewards than unpredicted rewards, and (3) a decrease in activity following reward omission. With respect to reward prediction, both VS and TS dopamine had a larger excitatory response to reward-predicting cues than cues predicting nothing (VS: p = $1.8 \times 10^{-10}$, TS: p = $6.5 \times 10^{-6}$, t-test, *Figure 6A, D*). Similarly, both VS and TS dopamine had a smaller response to a predicted reward than an unpredicted reward (VS: p = $5.1 \times 10^{-11}$, TS: p = $1.0 \times 10^{-4}$, t-test, *Figure 6A,D*). However, there was

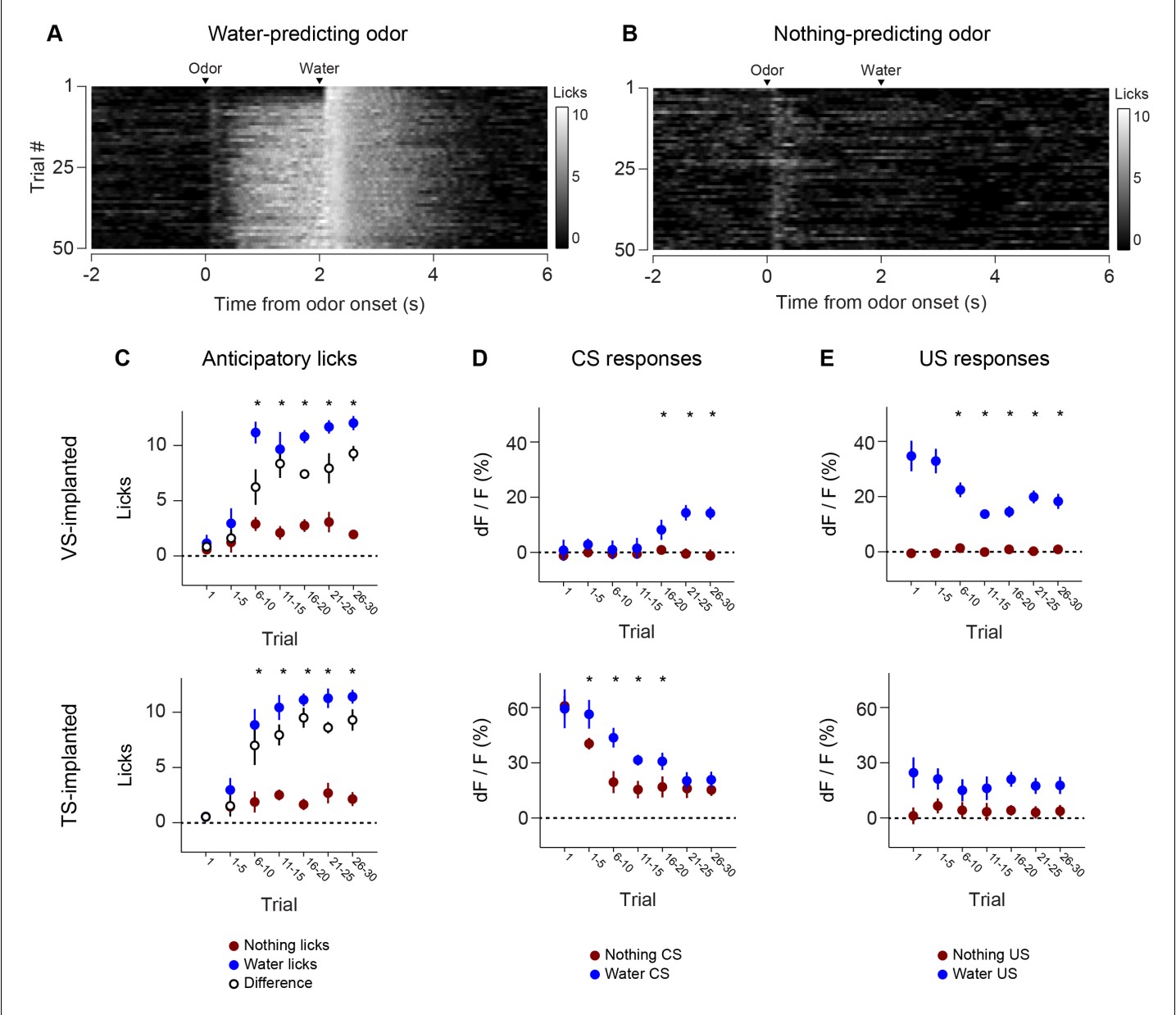

**Figure 5.** Dynamics of anticipatory licking behaviors and VS and TS dopamine. Licking in response to new odors predicting reward (**A**) or no outcome (**B**) over the course of a session, in animals that have been trained with many new odor associations, as in *Figure 4* (see Materials and methods). Separate plots for VS-implanted mice and TS-implanted mice are shown in *Figure 5—figure supplement 1*. (**C**) A quantification of the number of anticipatory licks elicited by each odor in VS-implanted animals (left) and TS-implanted animals (right). The difference between licks following a rewarding odor and an unrewarding odor are shown as open circles. (**D**) A comparison of the CS responses to rewarding and unrewarding new odors in VS dopamine (left) and TS dopamine (right). (**E**) A comparison of the US responses to either predicted water or predicted nothing in VS dopamine (left) and TS dopamine (right). The relationship between GCaMP responses in VS and TS and anticipatory licking is shown in *Figure 5—figure supplement 2*.

The following figure supplements are available for figure 5:

**Figure supplement 1.** Comparison of licking in VS-implanted and TS-implanted animals.

**Figure supplement 2.** Relationship between CS/US responses and anticipatory licking.

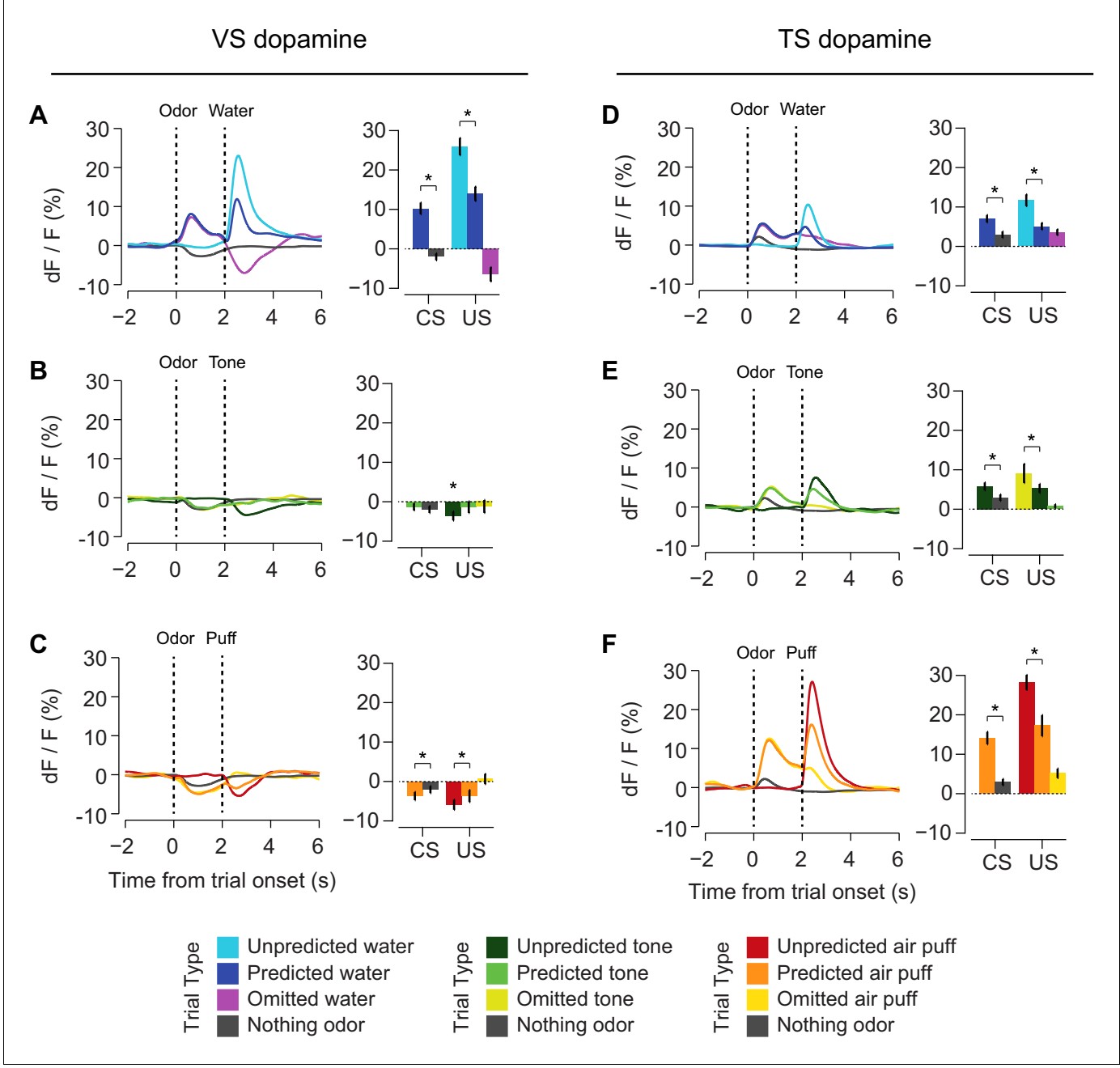

**Figure 6.** Responses to rewarding, aversive and neutral stimuli in VS and TS dopamine. Dopamine responses to water (**A**), tone (**B**), and air puff (**C**) in the ventral striatum and the posterior tail of the striatum (**D–F**). Plots of average traces from each region contain dotted lines indicating odor (CS) and outcome (US) delivery times. (**A, D**) Responses to unpredicted reward (cyan), predicted reward (blue), omitted reward (purple), and nothing odor (grey) are plotted in the left panels. For each trace, a quantification of the average peak response to the CS / US is shown on the right. (**B, E**) Responses to unpredicted tone (dark green), predicted tone (light green), omitted tone (yellow), and nothing odor (grey) are plotted in the left panels. For each trace, a quantification of the average peak response to the CS / US is shown on the right. (**C, F**) Responses to unpredicted air puff (red), predicted air puff (orange), omitted air puff (yellow), and nothing odor (grey) are plotted in the left panels. For each trace, a quantification of the average peak response to the CS / US is shown on the right. Data from individual sessions is shown in *Figure 6—figure supplement 2*. Behavioral responses to the tone are shown in *Figure 6—figure supplement 1*.

The following figure supplements are available for figure 6:

**Figure supplement 1.** Behavioral quantification of tone responses.
**Figure supplement 2.** Individual session data.

*Figure 6 continued on next page*

*Figure 6 continued*

a significant difference in the response to the omission of a predicted reward: whereas VS dopamine showed a dip below baseline following reward omission (VS: p = $4.8\times10^{-7}$, t-test, *Figure 6A,D*), TS dopamine axon signal was still significantly higher than baseline following omission (p = $2.2\times10^{-8}$, t-test, *Figure 6A,D*). In TS, reward prediction elicited sustained activity over the interval between odor presentation and reward onset, and reward delivery caused only a small increase above this level. In fact, although the average peak response was slightly higher in rewarded trials than unrewarded trials (p = 0.0014), the total response (area under each curve) after the outcome (reward delivery or omission) did not differ significantly (p = 0.81).

Interestingly, we found that the signals observed in TS dopamine displayed components of prediction error in response to non-rewarding stimuli as well. For example, TS dopamine showed less excitation to predicted air puff or tone than unpredicted air puff or tone (p = 0.00016, p = 0.00074, *Figure 6E–F*). Additionally, TS dopamine cue responses to air puff or tone predicting cues were larger than the responses to cues predicting no outcome (air puff: p = $6.8\times10^{-4}$, tone: p = $3.4\times10^{-4}$, *Figure 6E–F*). Similar to water omission, the omission of an expected tone or expected air puff did not cause a dip or increase in signal (*Figure 6E–F*).

In summary, VS dopamine encodes RPE whereas TS dopamine responds to salient stimuli in general. TS dopamine encodes the prediction of salient stimuli, and decreases the responses to salient stimuli once they are predicted, which are characteristics of prediction error. Notably, however, we did not observe clear responses to the omission of expected salient stimuli.

## Organization of dopamine novelty responses in VS and TS

Finally, we examined the relationship between different responses in dopamine and location in the striatum more carefully. For this purpose, in addition to VS and TS, dopamine axon signals in more anterior parts of the dorsal striatum (DMS and DLS) (*Figure 1B*) were recorded. Signals in each animal were pooled across sessions and the average was compared in relation to the location of recording sites (*Figure 7*). We first examined whether responses to novel odors were localized within VS or TS. VS consists of multiple sub-nuclei (*Zahm and Brog, 1992*) and it is suggested that there are functional differences between medial VS and lateral VS (*Ikemoto, 2007*). We did not observe systematic differences of novelty responses along dorsal-ventral or medial-lateral axis within VS (*Figure 7A*, anterior), although our spatial resolution could not completely distinguish each sub-nucleus. We did not observe noticeable differences between novelty responses in different sub-regions of TS either (*Figure 7A*, posterior).

We examined how novelty responses were localized in the striatum. The observed distribution supported the idea that dopamine novelty responses are localized in TS. Novelty responses were correlated with location along the anterior-posterior axis (r = −0.92, p = $1.68\times10^{-9}$, n = 40 animals, Pearson's correlation). We next examined responses to cues predicting rewarding, neutral or aversive outcomes, and responses to the omission of reward. Responses for all these factors were correlated with the location along anterior-posterior axis to various degrees (water: r = 0.61, p = $5.76\times10^{-5}$, nothing: r = −0.16, p = $2.61\times10^{-7}$, air puff: r = −0.76, p = $4.58\times10^{-8}$, and omission: r = −0.38, p = $1.68\times10^{-9}$, n = 40 animals). Finally, novelty responses were positively correlated with responses to nothing, air puff-predicting cues, and reward omission (nothing: p = $8.76\times10^{-7}$, air puff: p = $1.01\times10^{-14}$, omission: p = $9.02\times10^{-10}$, n = 40 animals). By contrast, water responses were observed in both VS and TS, although the amplitudes of water responses were slightly anti-correlated with novelty responses (p = 0.0138, n = 40 animals). We observed that water responses were found in all parts of the striatum (*Figure 7—figure supplement 2*).

Together, the differences in novelty responses and value coding between VS and TS dopamine suggested that novelty responses and value coding in dopamine axons are at least partially segregated in the striatum; lack of inhibitory responses to reward omission and excitatory responses to neutral or aversive stimulus are localized in TS and coincide with excitatory responses to novel stimuli.

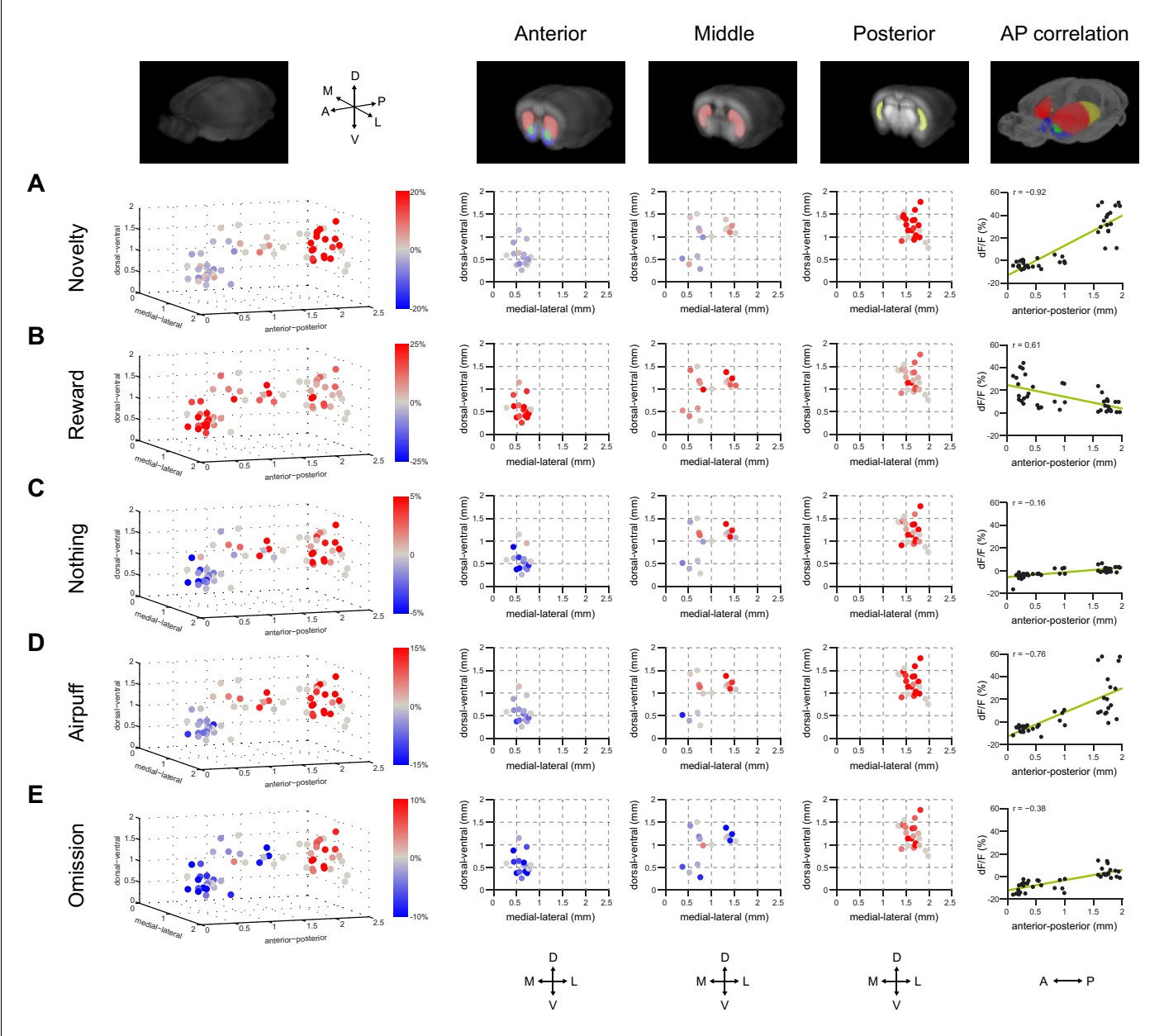

**Figure 7.** Maps of dopamine responses in VS and TS. The distribution of responses to novelty (A), reward (B), familiar odor predicting nothing (C), air puff (D), and reward omission (E). In the left panels, a 3D view of the average response from each animal. Novelty responses are the first responses to a novel odor either in the naïve case (i.e. *Figure 2*) or the trained case (i.e. *Figure 4*). Reward responses are the average response to unpredicted reward. Nothing responses are the response to cues predicting no outcome. Air puff responses are the average response to unpredicted air puff. Omission responses are the average response to the omission of expected water. In the middle panels, coronal max projections are shown from the 3D view. On the right, the correlation between signals from the fibers and their positions on the A-P axis is shown, along with a yellow line indicating the best fit. The plots of these responses are shown for VS, DMS, DLS, and TS in *Figure 7—figure supplement 2*. Examples of the whole-brain images used to find recording sites are shown in *Figure 7—figure supplement 1*.

The following figure supplements are available for figure 7:

**Figure supplement 1.** Examples of light-sheet images of cleared brains used to determine fiber locations.

**Figure supplement 2.** Responses to rewarding and aversive stimuli in VS, DMS, DLS, and TS.

## Discussion

In the present study, we examined dopamine novelty responses in different parts of the striatum. Our data demonstrate dramatic differences between dopamine axon signals in the ventral striatum (VS) (hereafter, VS dopamine) and dopamine axon signals in the tail of the striatum (TS) (hereafter, TS dopamine). Over the course of associative learning, TS dopamine and VS dopamine exhibited opposite initialization to novel cues. Large TS dopamine responses to novel cues gradually decayed, whereas VS dopamine had no response to novel cues and gradually developed responses to cues that predicted reward. Our previous study showed that dopamine neurons that project to the TS are anatomical outliers compared to many other dopamine neurons including dopamine neurons that project to the VS (*Menegas et al., 2015*). These new results suggest that dopamine neurons that project to TS are not only anatomically, but also functionally, distinct compared to dopamine neurons that project to VS.

### Opposite initialization to novel cues in VS and TS dopamine

The novelty responses we observed in TS dopamine, and especially the lack of novelty responses in VS dopamine, stand in stark contrast to popular theories explaining how novelty responses could fit into the dopamine RPE framework (*Hazy et al., 2010*; *Kakade and Dayan, 2002*). These theories proposed that dopamine neurons show 'optimistic initialization' to novel cues, promoting physical and/or computational exploration in search of potential rewards (often referred to as a 'novelty bonus'). However, our results suggest that the 'pure value'-coding VS dopamine axon signal does not include a novelty response, while the salience-coding TS dopamine axon signal does include a novelty response.

Over the course of classical conditioning, dopamine axon signals in VS and TS showed different coding principles. For a cue with an unknown outcome, VS dopamine initialized with no value prediction and gradually accumulated evidence for value, whereas TS dopamine began with a large excitation to novel cues and gradually decreased its responses. These dynamics can be conceptualized such that CS-value in VS dopamine was initialized with no value, while CS-salience in TS dopamine was initialized with very high salience.

The opposite dynamics in VS and TS dopamine during learning are reminiscent of two different views of associative learning: US associability and CS associability. Many models, including delta-rule, Rescorla-Wagner and temporal difference (TD) models (*Rescorla and Wagner, 1972*; *Sutton and Barto, 1998*; *Widrow and Hoff, 1960*), emphasize US associability; when an animal learns to associate a CS to a US, the nature of the *US* determines how well the animal can learn the association. In these models, US associability is determined by the prediction error of the US (i.e. the discrepancy between prediction and reality). Once the prediction error decreases, the US loses its effectiveness for creating an association with a CS. By contrast, other models (*Mackintosh, 1975*; *Pearce and Hall, 1980*) emphasize CS associability; when an animal learns to associate a CS to a US, the nature of the CS determines how well the animal can learn the association. In these models, CS associability alters attention to a stimulus, or the storage of a stimulus in working memory, to promote association with a US.

Dopamine dynamics in VS are well suited to serve as US associability, the teaching signal of the Rescorla-Wagner model (*Rescorla and Wagner, 1972*). TS dopamine does not follow the dynamics that the Rescola-Wagner model predicts. During the learning of novel odor-outcome associations, rather than US responses, CS responses were predominant and then decreased with learning. The dynamics of TS dopamine suggest that they may serve as CS associability, providing the attention signals for learning.

Novel stimuli may excite TS dopamine because of the unpredictability of the novel stimulus itself (i.e. it occurs with no cue or context predicting it) or because the outcome of these novel stimuli is unknown (i.e. they could predict a positive outcome, a neutral outcome, or a negative outcome). These two phenomena have been implemented in the framework of CS associability. For example, Wagner (*Wagner, 1978*) proposed that CS associability is correlated with the weakness of CS-context association (or the 'unpredictability' of the CS in the context). On the other hand, Pearce and Hall (*Pearce and Hall, 1980*) defined the CS associability as the unpredictability of the US (by the CS) in the previous trial. In either case, the novelty would promote learning because of high CS associability, although the latter model did not define CS associability on the very first trial. Our data

could be explained with either framework, but one observed phenomenon prefers the latter model. The responses to a novel odor decreased more slowly when the odor was associated with some outcome (water or air puff) than when it was associated with no outcome. If novelty responses are determined solely by odor-context association, responses to both odors should decrease at the same speed. Furthermore, this would lead to an equal response to all familiar odors, which we did not observe.

Importantly, previous learning models incorporated teaching signals in the Rescorla-Wagner model and attention signals by novelty responses in one dopamine RPE system, with novelty responses as an exception or a bonus to the system (*Hazy et al., 2010*; *Kakade and Dayan, 2002*). Here, we propose that CS associability, or the 'attention' term in dopamine signals may not be an exception in RPE, but instead may be a separate system – localized in particular brain regions such as in TS. Thus, dopamine in TS signals novelty and salience, and dopamine in VS signals RPE, although both systems may co-exist in some brain areas. In general, one big limitation of the current reinforcement learning algorithms is the so-called 'curse of dimensionality'. As the number of stimuli in an environment becomes large as in most natural environments, it quickly becomes difficult to properly assign credit to relevant objects. Attention would be critical to reduce the amount of information for learning to a more realistic amount. Dopamine in TS may be specialized for this function. In line with this idea, a recent study suggested that putative dopamine neurons in lateral SNc in monkey represented 'cognitive salience', which was correlated with working memory load (*Matsumoto and Takada, 2013*). Behaviorally, previous studies suggested that SNc is important for the acquisition of enhanced CS associability (*Lee et al., 2008*, *2006*). Mechanistically, dopamine in the prefrontal cortex has been modeled to serve a 'gating' function to provide flexible updating in working memory (*Braver and Cohen, 1999*), and a similar mechanism may apply to dopamine in TS. Of note, different from adjacent striatal areas, TS, categorized in caudal extreme (*Hintiryan et al., 2016*), does not receive projections from sensorimotor cortex, suggesting a functional distinction from other areas in the striatum. All in all, TS dopamine may function to 'pre-process' sensory inputs to weigh potentially important stimuli to make reinforcement learning more efficient in a complex environment.

Thus, during the learning of an association between a stimulus and a reward, dopamine signals in VS and TS may cooperate. Salience prediction error in TS dopamine may serve as the CS associability of the stimulus, whereas value prediction error in VS dopamine may serve as the US associability of the reward. Alternatively, similar to value prediction error in VS dopamine during stimulus-reward association, salience prediction error in TS dopamine may reinforce stimulus-stimulus associations.

## Different time-courses of CS and US responses in dopamine axons during learning

In contrast to TS dopamine, VS dopamine activity appeared to faithfully signal RPE. Several theories have arisen to explain how the reward prediction error could be computed in the brain. Of these, popular models such as Houk's model (*Houk and Adams, 1995*) and temporal difference (TD) models (*Sutton, 1988*; *Sutton and Barto, 1998*) assume that a single system controls both CS-related and US-related dopamine firing. Under these models, higher expectation causes CS responses to become larger and US responses to become smaller at the same time. However, we observed that changes in CS and US responses were not simultaneous. In VS, the decrease in response to predicted rewards (US) was faster, occurring over the course of a single session, whereas the increase in response to reward-predicting odors (CS) required weeks of training. Over-training accelerated the time course of these events, but not their temporal order.

The development of CS responses in VS dopamine was also much slower than the associative learning observed at the behavioral level. By contrast, US responses decreased as anticipatory licking increased. These results demonstrate that US responses in VS dopamine are well suited as prediction error signals in the Rescorla-Wagner learning model, whereas CS responses are not time-locked to this learning.

The time course of CS responses we observed in VS dopamine is not easily explained by simple TD models. In these models, during learning, RPE signals gradually transfer from the timing of the reward to the timing of the preceding stimulus (*Schultz et al., 1997*). Although the step-wise transfer may explain a delay between the decrease of US responses and emergence of CS responses, such a gradual transfer has not been observed in single neuron recording of dopamine neurons

(**Pan et al., 2005**). With any learning rate longer than one trial, transferred signals may become temporally smeared until they become time-locked to the CS (**Pan et al., 2005**; **Schultz et al., 1997**), which may not be detected in recording of single units. This theory would predict that we would observe some increase in signal between the stimulus and outcome (either smeared or sharp) during learning because the monitoring of population activity likely provides more reliable detection of small signals. However, this type of increase was not apparent with our bulk recording method. Instead, we observed a gradual development of a CS response directly following cue presentation. It is possible that distinct mechanisms could cause VS dopamine excitation in response to a reward-predicting CS and VS dopamine suppression in response to a predicted US, as proposed in several previous models (**Brown et al., 1999**; **O'Reilly et al., 2007**).

## Relation to previous studies

There are different types of novelty (**Schomaker and Meeter, 2015**). One example is spatial novelty, which could be signaled by different arrangements of objects/stimuli in the environment. This kind of environmental novelty is known to induce exploration of animals and accelerate learning in this environment (**Li et al., 2003**; **Otmakhova et al., 2013**). Another example is stimulus novelty, which is associated with objects/stimuli that animals have never encountered or do not remember. It has been reported that dopamine activity in VTA and dopamine in the ventral striatum increased in the former case, in novel environments (**Segovia et al., 2010**; **Takeuchi et al., 2016**). On the other hand, in the present study, we focused on the latter type of novelty. Dopamine axon responses to novel stimuli were localized in TS. One potential explanation is that depending on the training history and environments, a given type of novelty may cause the animals to expect potential rewards, resulting in the excitation of the value system. Interestingly, previous studies found that different brain areas are responsible for different kinds of novelty (**Schomaker and Meeter, 2015**). Further studies are needed to determine how dopamine in different striatal areas responds to different kinds of novelty in different training environments.

In our previous study, dopamine neurons that project to TS were mainly observed in the lateral SNc of mice (**Menegas et al., 2015**). A previous study proposed that putative dopamine neurons in the lateral SNc of monkeys encode 'motivational salience', which is the absolute value of positive or negative 'value' (**Matsumoto and Hikosaka, 2009**). On the other hand, another study proposed that excitation of dopamine neurons in response to non-rewarding stimuli encodes the stimulus intensity, regardless of value (**Fiorillo et al., 2013**). In the present study, the excitation of TS dopamine elicited by various neutral stimuli suggested that the responses in TS could be related to more general salience rather than motivational salience, although we cannot rule out the possibility that the tone and odor predicting nothing had positive or negative motivational values. On the other hand, the fact that signals encoded by TS dopamine are modulated by prediction suggests that they are not encoding pure physical salience (i.e. stimulus intensity). Instead, TS dopamine appears to encode general stimulus salience prediction error, which includes prediction-dependent suppression and prediction. The novelty responses we observed may be the extreme case of salience prediction error, causing large excitation because of minimum prediction, rather than an exception.

A recent study found that putative dopamine neurons which project to the tail of the caudate (part of posterior striatum) in monkey formed another group of dopamine neurons. These dopamine neurons did not respond to water reward but encoded 'sustained values' of visual cues, whereas putative dopamine neurons projecting to the anterior caudate encoded 'updating values', when cue-outcome contingency was frequently changed (**Kim et al., 2015**). However, our results indicate that the difference between VS and TS dopamine extends beyond their flexibility. The dynamics between them are different in nature, not only in learning speed. Most importantly, we found that TS dopamine did not encode 'values'.

Where do salience signals come from? How are salience signals regulated by novelty and experiences? A map of monosynaptic inputs to TS-projecting dopamine neurons should provide critical information (**Menegas et al., 2015**). Previous studies showed that various brain areas including olfactory and visual systems are modulated by experience (**Boehnke et al., 2011**; **Kato et al., 2012**). Whether dopamine neurons receive this processed information from sensory systems or whether more abstract information about salience and novelty is passed to dopamine neurons and sensory systems in parallel is an open question. Of note, behavioral responses to novel odors are very quick, within one respiration cycle in rats (**Wesson et al., 2008**). The responses to novel odors in TS

dopamine that we observed began within 200 ms, most likely within one respiration cycle, suggesting a potential contribution at the early stages of novelty.

## Technical considerations

Optical fiber fluorometry (fiber photometry) was developed by Kudo et al. (*Kudo et al., 1992*) and has been applied in many studies to record the population activity of neurons from cell bodies, dendrites, or axons (*Adelsberger et al., 2005*; *Davis and Schmidt, 2000*; *Murayama et al., 2007*). In this study, we recorded the population activity of dopamine axons in the striatum using GCaMP6m (*Akerboom et al., 2012*; *Chen et al., 2013*; *Kim et al., 2016*; *Parker et al., 2016*). We should point out several limitations associated with the present technique. First, previous studies (*Fiorillo et al., 2013*; *Schultz, 2015*) proposed that there is a temporal separation of two signals (stimulus intensity and value) in single dopamine neurons. However, we may only be able to measure the sum of these signals because of the limited temporal resolution of our method (population calcium imaging using GCaMP6m). Second, a recent study found that dopamine axons with distinct signals (locomotion and reward) coexist in the dorsal-most part of the dorsal striatum (*Howe and Dombeck, 2016*). Axons signaling different information might also co-exist in other areas of the striatum, and this could not be resolved with our bulk-imaging method (because such signals would effectively be 'averaged'). Third, because the spatial resolution of z-axis is large with fluorometry (~500 μm), we have to be careful in interpreting the analysis of differences along dorsal-ventral axis.

Dopamine axons passing through and below the ventral striatum to the cortex (*Aransay et al., 2014*) may have contributed to the signals in VS dopamine, although calcium transients in passing axons are smaller than in axon terminals and boutons (*Koester and Sakmann, 2000*; *Llano et al., 1997*). Finally, the activity of axons of dopamine neurons may not directly correspond to amounts of dopamine release at synapses or spike activities in cell bodies. Dopamine neurons that project to the ventral striatum (mainly medial shell) are able to co-release glutamate (*Stuber et al., 2010*). Neuronal activities can be modulated locally at axon terminals in the striatum by cholinergic neurons (*Threlfell et al., 2012*). Most importantly, observed calcium transients may not reflect spike counts, because of autofluorescence, bleaching, motion artifacts and inevitable normalization. Although we only applied baseline normalization (i.e. signals were subtracted with and then divided by the average signal in a 1 s period before CS in each trial) in this study, additional methods using activity-independent wavelength of excitation (*Kudo et al., 1992*; *Lerner et al., 2015*) or examination of emission spectrum (*Cui et al., 2013*) may improve fidelity, especially in freely moving animals.

## Conclusion

We found that dopamine responses to novel stimuli are more localized than previously believed. We propose to revise current RPE models so that novelty-driven and salience-driven attention is attributed to TS dopamine, rather than added to the RPE framework as a bonus (*Kakade and Dayan, 2002*). Thus, TS dopamine may be specialized for functions apart from value, such as attentional orientation (*Redgrave et al., 1999*), working attention, and/or as a filter for learning (*Braver and Cohen, 1999*; *Dayan and Sejnowski, 1996*; *Matsumoto and Takada, 2013*; *Pearce and Hall, 1980*). Further, our proposal includes another important point: RPE is not contaminated or distorted in VS dopamine. VS dopamine purely signals RPE, increasing the validity of the original ideas regarding dopamine's role in reinforcement learning (*Schultz et al., 1997*).

## Materials and methods

### Animals

85 male adult mice were used. These mice were the result of a cross between DAT (*Slc6a3*)-Cre mice (recombinase under the control of the dopamine transporter, DAT) (B6.SJL-*Slc6a3*$^{tm1.1(cre)Bkmn}$/J, Jackson Laboratory; RRID:IMSR_JAX:006660) (*Bäckman et al., 2006*) and tdTomato mice such that they were heterozygous for DAT-Cre and also heterozygous for tdTomato (*Gt(ROSA)26Sor*$^{tm9(CAGtd-Tomato)Hze}$, Jackson Laboratory). Animals were housed on a 12 hr dark/12 hr light cycle (dark from 07:00 to 19:00), one to a cage, and performed the task at the same time each day. All procedures were performed in accordance with the National Institutes of Health Guide for the Care and Use of Laboratory Animals and approved by the Harvard Animal Care and Use Committee.

## Viral injections, fiber implants, and head-plate installation

To prepare animals for recording, we performed a single surgery with three key components: (1) AAV-FLEX-GCaMP virus infection into the midbrain, (2) head-plate installation, and (3) one or more optic fiber implants into the striatum. At the time of surgery, all mice were 2–3 months old. All surgeries were performed under aseptic conditions with animals anesthetized with isoflurane (1–2% at 0.5–1.0 l/min). Analgesia (ketoprofen, 5 mg/kg, I.P.; buprenorphine, 0.1 mg/kg, I.P.) was administered for 3 days following each surgery.

To express GCaMP specifically in dopamine neurons, we unilaterally injected 250 nl of AAV5-CAG-FLEX-GCaMP6m ($1 \times 10^{12}$ particles/ml, Penn Vector Core) into both the VTA and SNc (500 nl total). To target the VTA, we injected virus at Bregma −3.0, Lateral 0.6, at all depths between 4.5 and 4.0 mm. To target SNc, we injected virus at Bregma −3.3, Lateral 1.6, at all depths between 4.0 and 3.5 mm. Virus injection lasted several minutes, and then the injection pipette was slowly removed over the course of several minutes to prevent damage to the tissue.

So that mice could be head-fixed during recording, we installed a head-plate onto each mouse. To do this, we removed the skin above the surface of the brain, dried the skull using air, and glued the head-plate onto the top of the skull with C and B Metabond adhesive cement. We used circular head-plates to ensure that the skull above the striatum would be accessible for fiber implants. Finally, during the same surgery, we also implanted optic fibers into the VS, DMS, DLS, and TS (1–4 fibers per mouse). To do this, we first slowly lowered optical fibers (either 200 μm or 400 μm diameter, Doric Lenses) into the striatum. The coordinates we used for targeting were as follows: (VS) Bregma 1.0, Lateral 1.25, Depth 4.1, (DS) Bregma 0.0, Lateral 1.5, Depth 2.25, (DLS) Bregma −0.5, Lateral 2.75, Depth 2.5, (TS) Bregma −2.0, Lateral 3.25, Depth 2.5. Once fibers were lowered, we first attached them to the skull with UV-curing epoxy (Thorlabs, NOA81), and then a layer of black Ortho-Jet dental adhesive (Lang Dental). After waiting fifteen minutes for this glue to dry, we applied a very small amount of rapid-curing epoxy (Devcon, A00254) to attach the fiber cannulas even more firmly to the underlying glue and head-plate. After waiting fifteen minutes for the epoxy to dry, the surgery was complete.

## Fiber fluorometry

Fiber fluorometry (photometry) (*Kudo et al., 1992*) allows for recording of the activity of genetically defined neural populations in behaving mice by expressing a genetically encoded $Ca^{2+}$ indicator, GCaMP6m (*Akerboom et al., 2012*; *Chen et al., 2013*) and chronically implanting an optic fiber. The optic fiber (200 μm or 400 μm diameter, Doric Lenses) allows chronic, stable, minimally disruptive access to deep brain regions and interfaces with a flexible patch cord (Doric Lenses) on the skull surface to simultaneously deliver excitation light (473 nm and 561 nm, Laserglow Technologies) and collect GCaMP and tdTomato fluorescence emission (see *Figure 1—figure supplement 1*).

Activity-dependent fluorescence emitted by cells in the vicinity of the implanted fiber's tip was spectrally separated from the excitation light using a dichroic, passed through a single band filter, and focused onto a photodetector connected to a current preamplifier (SR570, Stanford Research Systems). To record $Ca^{2+}$ transients from dopamine terminals, we injected a Cre-dependent adeno-associated virus (AAV) carrying the GCaMP6m gene into the VTA and SNc of transgenic DAT-Cre mice and implanted 200 μm or 400 μm diameter optic fibers in the striatum.

During recording, optic fibers were connected (1–2 per recording session) to patch cables which delivered excitation light (473 nm and 561 nm) and collected all emitted light. The emitted light was subsequently split and filtered (see *Figure 1—figure supplement 1*) and collected by a photodetector connected to a current preamplifier. This preamplifier output a voltage signal which was collected by a NIDAQ board. The NIDAQ board was connected to the same computer that was used to control odor, water, tone, and air puff delivery with Labview, so GCaMP and tdTomato signals could be readily aligned to task events such as odor delivery or reward delivery.

## Behavior

After surgery, mice were given three weeks to recover and become habituated to the installed head-plate and implanted optic fibers. Additionally, this allowed time for viral expression. After this recovery period, mice were handled for 2–3 days and water deprived. Weight was maintained above 90% of baseline body weight.

In the first 2–3 sessions, mice were head-fixed and given unexpected water at random intervals (randomly drawn, between 1 and 20 s, with a mean of 10 s and a normal distribution). This allowed mice to become habituated to being head-fixed and allowed us to determine the appropriate laser power (typically between 0.1 mW and 0.25 mW) to record >5% dF/F free water responses (typically between 10% and 50%). These sessions were important, so that recordings during odor-water association could begin from the very first odor presentation on the first day of data collection (see Experimental Timeline).

The volume of water was constant for all reward trials (predicted or unpredicted) in all conditions. Similarly, the same mild tone (15 kHz, 0.5 s, ~50 dB) was used in all tone trials and the same intensity air puff was used in all air puff trials. Each behavioral trial began with an odor cue (a puff of odor lasting 1 s), followed by a 1 s delay, and then an outcome (either water, nothing, tone, or air puff). Odors were delivered using a custom olfactometer (*Uchida and Mainen, 2003*). Each odor was dissolved in mineral oil at 1:10 dilution. 30 µl of diluted odor was placed inside a filter-paper housing (Thomas Scientific, Swedesboro, NJ). Example PID measurements are shown in *Figure 2—figure supplement 1*. Odors were selected pseudorandomly for each animal. Odorized air was further diluted with filtered air by 1:14 to produce a 1500 ml/min total flow rate. A variable inter-trial interval of 6–12 s (random) was placed between trials. All trial types were randomized in all of the sessions. Each day, the mice did about 300 trials over the course of about an hour. On a recording day, they performed the same task, and we recorded for ~45 min, which is approximately 250 trials, with constant excitation from the laser and continuous recording. Recordings from the same fiber were interspersed with at least two days of no recording.

## Experimental timeline

On the first day of classical conditioning, odors were presented to mice for the first time, and either predicted no outcome or reward. We quantified the 'novelty response' as the response to the first odor presentation that the mouse experienced, which was associated with no outcome (for n = 13 VS-implanted mice and n = 12 TS-implanted mice). For comparison, the response to the first unpredicted water presentation in those sessions was quantified as well. These 'novelty responses' were the first trials of the first day of classical conditioning (*Figure 2*), while the average responses for these sessions are reported as 'Day 1' and compared with later sessions in *Figure 3*. Due to technical difficulties we encountered in recording the first response of a session, some of the first responses were not recorded. Therefore, the sample size (number of animals) is lower for first trial responses (n = 12 mice for TS) in *Figure 2* than for average responses during 'Day 1' (n = 19 mice for TS) in *Figure 3*.

During classical conditioning, odor cues (also called 'conditioned stimuli' or 'CS') were associated with either reward or no outcome. In the case of reward trials, water (the 'unconditioned stimulus' or 'US') would follow odor presentation after 2 s, 90% of the time (i.e. 10% omission). In ~10% of trials, unpredicted water was delivered without odor presentation. During training, GCaMP responses were recorded at time points (*Figure 3*) rather than daily, to minimize bleaching or tissue damage. After 3 weeks of this classical conditioning training (with one water predicting odor, one nothing-predicting odor, and occasional unpredicted water), mice were introduced to new odor-outcome association types.

At this point in training, mice were also presented with odor-tone associations (20% of trials) or odor-air puff associations (20% of trials), in addition to the two familiar odors associated with water and with no outcome, allowing us to multiplex data from learning onto this data regarding value or salience coding and prediction error coding. Unpredicted tone or air puff was also delivered in ~5% of trials. Data from these sessions was used in *Figure 6* and *Figure 7*, including data from DMS-implanted or DLS-implanted mice.

Finally, a subset of these mice (n = 11) were trained with two new odors each day (one associated with water and one associated with no outcome), every day for a week, until mice began to discriminate between odors behaviorally within a few trials (see *Figure 5*) rather than over the course of many days (see *Figure 3*). We referred to these mice as 'overtrained mice'. After this training, we recorded GCaMP and licking signals from these overtrained mice as they learned either new odor-water (one third of sessions) or new odor-nothing associations (one third of sessions) in *Figure 4*. In one third of sessions, a new odor associated with air-puff was introduced in addition to the two familiar odors associated with water or with no outcome (*Figure 4—figure supplement 3*). We

randomized whether the new odor of a session would predict water, nothing, or air puff to ensure that mice could not generalize that novel odors reliably predicted a particular outcome. Mice performed one session per day.

## Fiber fluorometry and licking data analysis

GCaMP and tdTomato signals were collected as voltage measurements from the current preamplifiers using Labview (*Figure 1—figure supplement 1*). The 'dF/F' measurement was calculated by comparing the average signal in a 1 s period before each trial ('$F_1$') with the signal at any given point during the trial ('$F_2$'). The calculation for each point in the trial (calculated in 1 ms bins) was then simply $dF/F = (F_2 - F_1) / F_1$. We used this measurement because it readily normalized signals (i.e. in the case of low signal to noise ratio, the denominator would be larger). The average responses to a stimulus type within a session (often ~50 trials per stimulus type) were averaged, and these session averages were used as the data in each figure (individual session averages can be found in *Figure 3—figure supplement 1* and *Figure 6—figure supplement 1*, and example individual single trial traces are shown in *Figure 1* and *Figure 1—figure supplement 1*).

These session averages were compared across animals in two basic ways. (1) Traces were averaged and plotted (as the average of all session averages) along with the standard error (the total number of sessions being the sample size) as in *Figure 5*, left panels. (2) Peak responses to cues/outcomes were quantified by finding the point with the maximum absolute value during 2 s following cue/outcome for each trial, then comparing the averages between sessions as in *Figure 5*, right panels. Because traces were aligned using task events (i.e. cue on time) rather than behavioral events (i.e. first inhalation), comparing peak responses ensured that signals, which were slightly offset in time relative to odor presentation, could be compared.

To compute the latency of responses to novel cues, each trial was tested for difference from baseline in the first five novel odor trials using time bins of 50 ms. We called the 'latency' of the response (in each trial) the center of the first time bin where five consecutive time bins all showed significant difference from baseline. To compute the latency of novelty discrimination, the sessions were tested for significant difference between familiar and novel odors in the first five trials using time bin of 50 ms. We called the 'latency' of discrimination the center of the first time bin where five consecutive time bins all showed significance.

While recording GCaMP signals, we also recorded licking. To measure licking, we used a detector that output a voltage based on the disruption of its infra-red light path. We set a threshold for signal corresponding to a 'lick' and then made the signal binary by finding each time point where the signal crossed the threshold so that it could easily be quantified. Our main quantification for licking was counting the number of 'anticipatory licks', the licks following an odor (CS) and preceding the arrival of the outcome (US).

GCaMP responses and licking responses were collected through Labview during the training for offline analysis. Statistical analyses (i.e. t-tests, ANOVA) were run using Matlab (Mathworks). All analyses considered a value of $p \leq 0.05$ significant, and exact p-values are reported in the text. To quantify body movement, we used a video camera to capture images of the mice while they performed the task and made a rough estimate of total body movement by subtracting each frame of the video from the last frame, using the 'imabsdiff' function in Matlab. We reported these values, which we took to be a proxy for body movement, as 'arbitrary units' or 'a.u.' because they were measured in pixels.

## Tissue clearing using CLARITY

Brains were cleared as previously described (*Menegas et al., 2015*) at 37°C for 2 days, with a constant current of 1.2 amps. A Niagra 120 V (Grey Beard Pumps #316, Mt Holly Springs, PA, United States) pump was used to circulate clearing solution. A Precision Adjustable 60 V/5A power supply (Korad Technology #KA6005D, Shenzhen, China) was used to provide current. A 5-gallon plastic container (US Plastic #97,028, Lima, Ohio, United States) was used as a clearing solution reservoir and tubing was run though a second 5-gallon plastic container filled with water to cool the solution flowing through it. Chambers were constructed as previously described (*Chung and Deisseroth, 2013*) using a Nalgene chamber (Nalgene 2118–0002, Rochester, NY, United States) and platinum wire (Sigma-Aldrich 267228, St. Louis, MO, United States). Clearing was done in a room held at 37°C.

## Imaging using light-sheet microscope

Images were acquired with the Zeiss Z.1 Light-sheet microscope (Carl Zeiss, Jena, Germany). Brains were glued to the tip of a 1 ml syringe (without needle) such that the posterior tip of the cerebellum was in contact with the syringe. A 488 nm laser was used to excite GFP and a 647 nm laser was used to produce autofluorescence. Images were collected through a 5× objective with PCO-Edge scMOS 16 bit cameras (PCO, Kelheim, Germany) with 1920 × 1920 pixels. Each frame was 2000 × 2000 µm, so each pixel was roughly 1.04 µm. The step size between images was set to 5.25 µm, so the voxels were not isotropic. Brains were imaged horizontally from the dorsal side, and then rotated 180° for horizontal imaging from the ventral side. Each view was tiled with 7 × 6 tiles (14,000 × 12,000 µm) and the two views were combined to create a continuous image. Autofluorescence images were subsequently downsized to 1400 × 1200 × 700 pixels for alignment to the reference space. In these downsized images, voxels have 10 µm spacing in all three dimensions. Brains were aligned to a previously described reference space comprised of the average of 25 brains (*Menegas et al., 2015*). Alignment to this reference space was performed using Elastix (*Klein et al., 2010*). We performed affine alignment followed by B-spline alignment based on mutual information, as previously proposed for human magnetic resonance imaging (MRI) image registration (*Metz et al., 2011*). After alignment, fiber positions were manually determined by tracing the fiber paths to their termination points.

## Classification of implant sites

After clearing and imaging each brain as a whole volume and aligning these images to determine the exact location of each fiber, we classified each fiber as either (1) an implant into VS, (2) an implant into DS, (3) an implant into TS, or (4) an incorrectly targeted fiber. Our analysis in the current paper focuses on comparing VS-implanted fibers to TS-implanted fibers. Most implants were successfully targeted to VS or TS. We classified implants as successful based on the following criteria. For VS: any fiber within the nucleus accumbens core or shell, between Bregma 2.0 and Bregma 0. For DS: any fiber within the striatum anterior to Bregma 1.5. For TS: any fiber within the striatum posterior to Bregma −1.5. We discarded data from eight animals which had fibers incorrectly targeted. The fibers in these animals were often in areas of cortex directly adjacent to the intended recording site. We observed very little or no signal (compared to our other recordings) in these cases, likely due to the relatively sparse dopaminergic innervation of cortex relative to striatum in mouse.

## Acknowledgements

We would like to thank members of the Uchida lab for comments on the manuscript and helpful discussions. We thank S Matias, Z Mainen (Champalimaud Institute of Unknown) and C Burgess, M Andermann (Harvard Medical School) for advice on fiber fluorometry. Additionally, we would like to thank the staff of the Harvard Center for Biological Imaging, Edward Soucy (CBS neuro-engineering platform), and Louis Baum (Harvard physics department) for technical support. Finally, we would like to also thank Vivek Jayaraman, PhD, Rex A Kerr, PhD, Douglas S Kim, PhD, Loren L Looger, PhD, and Karel Svoboda, PhD from the GENIE Project, Janelia Farm Research Campus, Howard Hughes Medical Institute for AAV-GCaMP6m. This work was supported by National Institutes of Health grants R01MH095953 (NU), R01MH101207 (NU), R01MH110404 (NU), Harvard Mind Brain and Behavior faculty grant (NU) and Fondation pour la Recherche Medicale grant SPE20150331860 (BB).

## Additional information

### Competing interests

NU: Reviewing editor, *eLife*. The other authors declare that no competing interests exist.

### Funding

| Funder | Grant reference number | Author |
|---|---|---|
| National Institute of Mental Health | R01MH095953 | Naoshige Uchida |

| | | |
|---|---|---|
| Harvard Mind Brain and Behavior | | Naoshige Uchida |
| National Institute of Mental Health | R01MH101207 | Naoshige Uchida |
| National Institute of Mental Health | R01MH110404 | Naoshige Uchida |
| Fondation pour la Recherche Médicale | SPE20150331860 | Benedicte M Babayan |

The funders had no role in study design, data collection and interpretation, or the decision to submit the work for publication.

### Author contributions

WM, Conceptualization, Data curation, Formal analysis, Writing—original draft, Writing—review and editing; BMB, Methodology, Writing—review and editing; NU, Supervision, Writing—review and editing; MW-U, Conceptualization, Supervision, Writing—original draft, Writing—review and editing

### Author ORCIDs

Naoshige Uchida, http://orcid.org/0000-0002-5755-9409
Mitsuko Watabe-Uchida, http://orcid.org/0000-0001-7864-754X

### Ethics

Animal experimentation: This study was performed in strict accordance with the recommendations in the Guide for the Care and Use of Laboratory Animals of the National Institutes of Health. All of the animals were handled according to approved Harvard animal care and use committee (IACUC) protocols (#26-03) of Harvard University. All surgery was performed under isofluorane anesthesia, and every effort was made to minimize suffering.

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
