## [Decision Letter]

Thank you for submitting your article "Opposite initialization to novel cues in dopamine signaling in ventral and posterior striatum" for consideration by *eLife*. Your article has been reviewed by two peer reviewers, and the evaluation was overseen by Gary Westbrook as the Senior Editor. The following individual involved in review of your submission has agreed to reveal her identity: Kay M Tye (Reviewer #2).

Both reviewers were very positive about your work. The reviewers have discussed the reviews with one another and the Senior Editor has drafted a consolidated summary of the reviews to help you prepare a revised submission.

Summary:

Menegas and colleagues studied dopamine neuron responses during classical conditioning and compare the responses across different striatal subregions. They used fiber photometry to record calcium responses from DA axons and tracked these signals across learning. The major result is a dramatic difference between dopamine signals in the ventral striatum (VS) compared to the tail of the striatum (TS). While VS projecting DA neurons followed the classical reward prediction error patterns and did not respond to novel cues, TS projecting DA neurons responded strongly to novel cues and also to the overall salience of cues and outcomes. This is a beautiful manuscript reporting on a very timely and important topic. The design of the experiments is elegant and the observed difference between DA responses is striking. The manuscript is very well written and describes an important finding with attention to the points listed below.

Essential revisions:

1) It would be important to measure the timecourse of the odor signals to assess whether cue responses in the TS are concordant with what is expected based on adaptation in the olfactory bulb or piriform cortex. Please use PID measurements of olfactometer output. This would also inform the issue of whether TS implements a novelty filter or rather it inherits novelty signals.

2) Why did the authors use TdTomato as the control instead of the 405 emission? Did the authors also collect this? Do the authors have some evidence that different emission wavelengths produce identical mechanical deflections, or might there be chromatic aberrations that make this different? At this point, the appropriate controls for fiber photometry are still up for discussion, so it would be useful to have the authors directly discuss this issue.

3) The description of TS signals beyond novelty and their relationship to RPE is confusing. Would it be correct to simplify the responses as the absolute value of RPE plus a novelty bonus? Either way, please clarify.

4) It was not convincing that the neutral tone stimuli (Figure 6) are perceived as truly neutral. Why then do they provoke negative inflections in the RPE-type signal observed in VS? Perhaps video tracking of startle response would help to exclude this possibility.

5) The authors should not discuss the signals as "dopamine signals" as they are only looking at the calcium transients within dopamine neurons that may co-release glutamate. They should instead refer to these signals as "signals from VTA DAT+ neurons in the VS or TS" or something similar, and separately discuss what this can reflect.

6) It would be important to see histology showing representative expression of GCaMP-expressing fibers in each of the regions sampled

7) How much of the photometry signal is due to terminals versus axons of passage?

8) Regarding Figure 5: What happens if the authors sort the trials where there was licking versus no licking (in water trials) in terms of the response to the water delivery. No licking may indicate that the mouse did not recognize/detect/attend to the odor cue?

9) It would be nice to see the anatomy: if the authors retrogradely label VTA DA neurons that project to VS versus TS, where do these TS neurons reside? What is the distribution? This would be helpful information in terms of joining this information with the work by Stephan Lammel on the arrangement of VTA DA neurons projecting to mPFC or NAc m shell or NAc l shell.

10) Did the authors control for the order of all of the manipulations? Were the order of manipulations randomized for both the trial types within A, B, C and the epochs (particularly in Figure 6)?

---

## [Author Response]

*Essential revisions:*

*1) It would be important to measure the timecourse of the odor signals to assess whether cue responses in the TS are concordant with what is expected based on adaptation in the olfactory bulb or piriform cortex. Please use PID measurements of olfactometer output. This would also inform the issue of whether TS implements a novelty filter or rather it inherits novelty signals.*

These are very important points, and we appreciate this advice. During odor-association learning, we observed that novel-odor-cue-evoked calcium signals in TS-projecting dopamine axons decreased within a session (Figure 2 for initial learning and Figure 4 for repeated learning), as well as over days (Figure 3).

One potential explanation for the within-session decrease in signal could be that the effective concentration of the odorants decreased over time within a session, and that the TS dopamine axon signals encode stimulus intensity and merely reflect the concentration change. To test this idea, we quantified the change in the effective concentration of the odorants used in our study over trials within a session, as well as the latency of odor delivery within single trials, using a photoionization detector (PID) (Aurora Scientific, 200B: miniPID Dispersion Sensor) (Figure 2—figure supplement 1). We found that some odors were slow to decay (Figure 2—figure supplement 1), and others decayed significantly over the course of a session (Figure 2—figure supplement 1). Therefore, we split our odors into two groups, based on whether they decayed slowly or quickly (Figure 2—figure supplement 1). We then compared TS dopamine axon responses to novel odor cues based on the stability of the odorant over time. We found that responses to both types of odors decreased dramatically over the course of a session (Figure 2—figure supplement 1). Another way to address this potential concern is to look at TS dopamine axon responses to a familiar odor over the course of a session. We performed this analysis and found that responses to familiar cues were stable over the course of a session (Figure 2—figure supplement 2). Finally, we also observed a decrease in responses to novel odors across days (Figure 3), which would not be affected by any within-session decay of odor concentration. Taken together, we do not think that changes in odorant concentrations affected our main conclusions. We added sentences to the Results section, as specified below.

“To determine whether this decrease in signal was caused by a decrease in effective odor concentration within sessions, we compared responses to fast-decaying and slow-decaying odors (Figure 2—figure supplement 1). TS dopamine responses to novel odors, both fast-decaying and slow-decaying odors, decreased over the course of a session (Figure 2—figure supplement 1), whereas TS dopamine responses to familiar odors did not change (Figure 2—figure supplement 2).”

Whether the decrease of TS dopamine axon responses to novel cues is due to sensory adaptation is also a fundamental question meriting some discussion. First, studies using anesthetized animals have shown that neurons in the piriform cortex, but not in the olfactory bulb, quickly adapt to repeated odor presentations, and recover in a relatively short time period, normally within tens of seconds (Wilson 1998, Best and Wilson 2004). Our observation that dopamine novelty response decreases over days (Figure 3) cannot be explained simply by this type of adaptation. In addition, the decay in response was unique to novel odors and we did not observe similar decay in signal elicited by familiar odors (Figure 2—figure supplement 2). Further, the speed of decrease in responses to novel odors depends on whether the odor predicted nothing, water, or air puff. We observed that TS dopamine axon responses to novel odors decreased more slowly when the novel odor predicted water (Figure 5) or airpuff (Figure 4—figure supplement 2) than when it did not predict any outcome. It is unlikely that these differences in the decay time course can be explained simply by sensory adaptation.

A recent study suggested that neurons in the olfactory bulb of awake mice (Kato et al., 2012) modulated their responses to the repetitive exposure of odors following more complicated principles than the simple adaptation observed in olfactory sensory neurons (Chaput, 2000). Kato et al. showed that neurons in the olfactory bulb quickly reduced their responses to odors following repetitive exposure (within a session) and that the effects of exposure lasted for over a month. Although we cannot directly compare the time-course because of the difference of tasks (i.e. the odor presentations are interspersed with other trial types in our task, rather than repeated consecutively as in their task), the dynamics in olfactory bulb are comparable to our observation in TS dopamine axon signals.

It is also important to discuss the latency of novelty detection within a trial. Behaviorally, novelty detection occurs within 200 ms; immediately after a single sniff (Wesson et al., 2008). In order to understand which brain area could function as a novelty filter, as the reviewers mentioned, it is very important to examine the temporal dynamics of neuronal signals in response to novelty. We found that the median latency of TS dopamine axon responses to novel odors in naïve mice was 140 ms (Figure 2—figure supplement 4) and the median latency of discrimination between novel and familiar odor in trained mice was 170 ms (Figure 4—figure supplement 3). We plotted the average of the responses to the first 5 presentations of a novel odor in 12 mice (Figure 2—figure supplement 4). The short latency of these responses means that the signal is likely to arise within the first sniff, based on a previously reported sniff rate of 3.4 Hz (294ms/cycle) (Kato et al., 2012). Because we used GCaMP, a calcium-based indicator, our temporal resolution is somewhat limited. This is also an issue with the previous studies that we mentioned (Kato et al., 2012, Wesson et al., 2008). Notably, the peak of the average response we observed in TS dopamine axon responses to novel odors was not until 600 ms after odor delivery (Figure 2—figure supplement 4).

In order to understand the precise temporal relationship between these signals, functional analysis and more precise measurements of timing (single unit and simultaneous awake recording), as well as consideration of temporal coding in the olfactory bulb (Cury and Uchida, 2010) is necessary. In short, our results demonstrate that TS dopamine axons signal novelty quickly (within the first sniff), but future studies will examine which brain area or areas function as novelty filters. We added a brief paragraph to the Discussion (subsection “Relation to previous studies”) to address this issue.

“Where do salience signals come from? How are salience signals regulated by novelty and experiences? A map of monosynaptic inputs to TS-projecting dopamine neurons should provide critical information (Menegas et al., 2015). Previous studies showed that various brain areas including olfactory and visual systems are modulated by experience (Boehnke et al., 2011; Kato et al., 2012). Whether dopamine neurons receive this processed information from sensory systems or whether more abstract information about salience and novelty is passed to dopamine neurons and sensory systems in parallel is an open question. Of note, behavioral responses to novel odors are very quick, within one sniff in rats (Wesson et al., 2008). The responses to novel odors in TS dopamine that we observed began within 200 ms, most likely within one sniff, suggesting a potential contribution at the early stages of novelty.”

*2) Why did the authors use TdTomato as the control instead of the 405 emission? Did the authors also collect this? Do the authors have some evidence that different emission wavelengths produce identical mechanical deflections, or might there be chromatic aberrations that make this different? At this point, the appropriate controls for fiber photometry are still up for discussion, so it would be useful to have the authors directly discuss this issue.*

Optical fiber fluorometry has long been used in various preparations (Kudo et al., 1992, Duff Davis and Schmidt, 2000, Adelsberger et al., 2005, Murayama et al., 2007). Signals collected from excitation with different wavelengths are sometimes used to normalize signals with the goal of removing the effects of autofluorescence, bleaching, and movement (Kudo et al., 1992, Lerner et al., 2015). This is an especially important consideration in freely moving animals, where the interface between the implanted fiber and the patch cord used to collect signal can cause additional artifacts. In this paper, we used a head-fixed preparation and reported the signal from GCaMP. We did not normalize by the tdTomato signal because we were not confident that the ratio of GCaMP: tdTomato would be constant between mice. Specifically, TdTomato levels were likely to be similar (because this was genetically encoded), but the virally-delivered GCaMP was likely to be present at slightly different levels in each animal. This problem would not likely be solved by using 405 nm excitation to separately collect a baseline as previously outlined (Lerner et al., 2015) because this autofluorescence signal would also not scale with GCaMP infection size. We added sentences in the Discussion (“Technical considerations”).

“Most importantly, calcium transients may not enhance spike counts, because of autofluorescence, bleaching, motion artifacts and inevitable normalization. Although we only applied baseline normalization in this study, additional methods using activity-independent wavelength of excitation (Kudo et al., 1992; Lerner et al., 2015) or examination of emission spectrum (Cui et al., 2013) may improve fidelity, especially in freely moving animals.”

In order to avoid drift in signals due to bleaching, we minimized laser power for each mouse (approximately 0.25 mW, subject to slight fluctuations in power between days) and ensured that our recording was limited to approximately 45 min per day. Additionally, although we trained mice every day, we recorded only at time points during training (Figure 3). We found that responses to free water did not change significantly within sessions in VS dopamine axons (Figure 2—figure supplement 2) or TS dopamine axons (Figure 2—figure supplement 2). We added this to our Results section as well (Results “Excitation to novel cues in TS dopamine”).

“To determine whether changes in signal intensity observed within a session were likely to have been related to bleaching, we compared responses to free water and familiar odors over the course of sessions (Figure 2—figure supplement 2). We found that both VS dopamine responses (Figure 2—figure supplement 2) and TS dopamine responses (Figure 2—figure supplement 2) to free water remained constant within sessions. Similarly, TS dopamine responses to familiar odors predicting no reward remained constant as well (Figure 2—figure supplement 2).”

To determine whether our signals were likely to have been contaminated by movement-related artifacts, we collected video of mice (n=6) performing the task and analyzed body movement (Figure 2—figure supplement 3). We quantified total body movement from all regions of the mouse, and found that they were not time-locked with novel (Figure 2—figure supplement 3) or familiar (Figure 2—figure supplement 3) odor presentation. However, we did find that there was body movement time-locked to the delivery of odors predicting reward (Figure 2—figure supplement 3). This is not surprising, because mice often perform a type of approach behavior (taking a few steps) when they expect reward, despite being head-fixed. We added a paragraph regarding motion artifacts to the results (Results “Excitation to novel cues in TS dopamine”). Moreover, the tdTomato traces in Figure 1 show that in the case of free water trials, when mice start licking at water delivery, no change in signal was observed in the tdTomato signal. Taken together, our analysis does not support the idea that the signals we discuss in the manuscript are contaminated by movement artifacts (gross body movements or licking).

“To determine whether the signals we observed could have been caused by a movement-related artifact, we used video analysis to quantify the total body movements of mice performing the task (Figure 2—figure supplement 3). We found that mice did not show gross body movements in response to odors predicting no outcome (Figure 2—figure supplement 3) or novel odors (Figure 2—figure supplement 3), although they performed a stereotypical approach behavior in response to odors predicting reward (Figure 2—figure supplement 3).”

*3) The description of TS signals beyond novelty and their relationship to RPE is confusing. Would it be correct to simplify the responses as the absolute value of RPE plus a novelty bonus? Either way, please clarify.*

There is a controversy about the excitation of dopamine neurons in response to aversive stimuli, and we therefore tried to avoid oversimplifying our results.

Hikosaka and colleagues (Matsumoto and Hikosaka, 2009) found that some dopamine neurons are activated by both aversive and appetitive stimuli. They interpreted this result as dopamine neurons encoding “motivational salience” (the absolute value of RPE). In this definition, it is the “aversiveness” or “negative value” of the stimulus that makes these stimuli motivationally salient. However, Fiorillo (Fiorillo et al., 2013) acquired a very similar dataset, but proposed that the excitation of dopamine neurons was better explained by stimulus intensity. He argued that this excitation was not correlated with aversion, which he estimated based on choice preferences. Unfortunately, our task was not designed to distinguish between these possibilities. Nonetheless, our data do not perfectly match with either of the ideas. First, we observed similar amplitudes of excitation in response to water and a very quiet tone in thirsty mice. For comparison to other studies, a 90 dB tone did not affect the utility function in thirsty monkeys (Fiorillo et al., 2013), and we used a 55 dB tone with our thirsty mice. Second, we also observed large responses to novel cues. We think that it is important to distinguish these TS dopamine axon signals from those proposed in previous studies. We therefore propose that signals in TS are general salience prediction error signals (containing both salience and novelty) rather than conceptualizing them as the absolute value of salience plus a separate bonus for novelty.

We cannot formally exclude other possibilities regarding TS dopamine axon signals. For example, if novel odors and mild tones are both aversive, TS dopamine axons could be signaling the absolute value of RPE. Or, as a reviewer suggested, responses to novel cues might be differentially conceptualized as a “bonus” added to the signal rather than an aspect of salience. We rephrased parts of our Results and Discussion to try to make our proposals clearer (Discussion “Relation to previous studies”).

“A previous study proposed that putative dopamine neurons in the lateral SNc of monkeys encode ‘motivational salience’, which is the absolute value of positive or negative ‘value’ (Matsumoto and Hikosaka, 2009). On the other hand, another study proposed that excitation of dopamine neurons in response to non-rewarding stimuli encodes stimulus intensity, regardless of value (Fiorillo et al., 2013). In the present study, the excitation of TS dopamine elicited by various neutral stimuli suggested that the responses in TS could be related to more general salience rather than motivational salience, although we cannot rule out the possibility that the tone and odor predicting nothing had positive or negative motivational values. On the other hand, the fact that signals encoded by TS dopamine are modulated by prediction suggests that they are not encoding pure physical salience (i.e. stimulus intensity). Instead, TS dopamine appears to encode general stimulus salience prediction error, which includes prediction-dependent suppression and prediction. The novelty responses we observed may be the extreme case of salience prediction error, causing large excitation because of minimum prediction, rather than an exception.”

4) It was not convincing that the neutral tone stimuli (Figure 6) are perceived as truly neutral. Why then do they provoke negative inflections in the RPE-type signal observed in VS? Perhaps video tracking of startle response would help to exclude this possibility.

To address this concern, we have performed behavioral measurements of responses to tones of various sizes. We found that there was not a large behavioral response to the size of tone that we used. However, we cannot say with certainty that the tone was not slightly aversive, so we changed our language in the text from “neutral tone” to “mild tone” or “quiet tone”.

To try to minimize the aversiveness of the tone, we used a very small 55 dB tone for this experiment. For reference, ambient noise in the room where recordings were performed was approximately 50 dB. Startle responses have been observed in response to very loud tones, in experiments with background tones that were louder than the tone that we chose (105 dB startle stimulus with 65-80 dB background with monkeys, Schneider et al., 2013, 139 dB with 70 dB background with rats, Hoffman and Searle, 1967, 110 dB with mice, Ouagazzal et al., 2001, 118 dB startle stimulus with 72 dB background with mice, Sallinen et al., 1998). It should also be noted that startle experiments could also actually have a component of perceptual salience (or, “surprise”), rather than pure aversion.

Because we used a tone that was not much louder than the background noise in the room, and because mice became familiar with this tone, we do not think that it was perceived as aversive. To test this, we measured the behavioral responses of mice when presented with tones of different volumes: 55 dB to 105 dB (Figure 6—figure supplement 1). We found that mice did not generally respond to smaller tones (Figure 6—figure supplement 1), and a louder tone was required to cause mice to freeze or remain motionless (Figure 6—figure supplement 1). We plotted the relationship between tone intensity and motion, and found that freezing occurred in response to tones larger than approximately 80 dB in our condition (Figure 6—figure supplement 1). Again, for comparison, the tone that we used in experiments was only 55 dB – and mice did not tend to freeze in response to this tone.

Fiorillo examined the effect of a louder tone (90 dB) on the utility function in a water-reward-choice tasks, and showed that their tone did not cause aversion in monkeys (Fiorillo et al., 2013). This means that the sound did not bias the monkey’s choice such that it would prefer an equally valuable option less often to prevent the sound from being played. Based on these (admittedly indirect) pieces of evidence, we suggest that signals in TS are likely to encode general salience rather than aversion or the absolute value of “value”. However, we cannot formally exclude the possibility that the 55 dB tone we chose was aversive to mice. Therefore, we rephrased the related sections of the paper. Specifically, we replaced “neutral” to “non-rewarding” in the Results and added a discussion on the behavioral responses we observed to this tone (Results section). Please also see Essential Revision #3.

“Responses to rewarding and non-rewarding stimuli in VS and TS dopamine

After several weeks of this training, in some sessions, trials with odors predicting either a mild tone (55 dB) or an air puff were interleaved in addition to trials with water and trials with no outcome. We chose a very mild tone with a similar intensity to the background noise in the room to try to minimize the aversiveness of this stimulus. To estimate the aversiveness of auditory stimuli, we measured the behavioral responses to tones of different volumes in a different set of mice (Figure 6—figure supplement 1). We found that quiet tones did not cause freezing.”

The question of why we observed decreases in VS dopamine in response to tone is interesting, and likely to be related. Our tasks include a high probability of reward (~50% of trials are rewarded). In such high reward contexts, neutral stimuli might be interpreted as a signal for “no reward” because they predict that reward will not come in the very near future (Fiorillo, 2013, Matsumoto et al., 2016). Because of this, we have to be careful when interpreting inhibition in VS caused by tone-predicting odor cues or by tones themselves. Notably, familiar odors predicting no outcome also caused a negative inflection in the RPE-type signal that we observed in VS that was similar in size (Figure 6).

*5) The authors should not discuss the signals as "dopamine signals" as they are only looking at the calcium transients within dopamine neurons that may co-release glutamate. They should instead refer to these signals as "signals from VTA DAT+ neurons in the VS or TS" or something similar, and separately discuss what this can reflect.*

We agree with the reviewer. We changed our wording in many parts of the paper (Abstract, Introduction and other places) to make it clear that the signals we observed are due to increases in calcium in the axons of midbrain dopamine neurons (DAT+) found in either VS or TS. We added a detailed explanation of our nomenclature in the Introduction and at the beginning of the Results section and added some discussion of the meaning of these axon signals in the Discussion section (Introduction, Results “Recording activity from dopamine axons in the striatum”, Discussion “Technical considerations”).

“We will call the bulk calcium signal that we observed from the axons of DAT+ midbrain dopamine neurons in the striatum “VS dopamine” and “TS dopamine” in the following sections”.

“We used optical fiber fluorometry (fiber photometry) to record bulk calcium signals from the axons of midbrain dopamine neurons projecting to several regions of the striatum (Kim et al., 2016; Kudo et al., 1992; Parker et al., 2016).”

“In this study, we recorded the population activity of dopamine axons in the striatum using GCaMP6m (Chen et al., 2013; Akerboom et al., 2012; Kim et al., 2016; Parker et al., 2016).”

*6) It would be important to see histology showing representative expression of GCaMP-expressing fibers in each of the regions sampled*

We added images of GCAMP-expressing fibers throughout the striatum in Figure 1—figure supplement 3. Specifically, we showed representative images from an animal with no virus injection (first row), VTA virus injection (second row), SNC virus injection (third row), and VTA+SNC virus injection (fourth row). All data in the paper was collected from animals with injections into both VTA and SNC.

*7) How much of the photometry signal is due to terminals versus axons of passage?*

It is difficult to exclude the possibility of having collected signals from passing axons, we estimated that most of the signals we observed come from axon terminals based on four arguments to this effect.

First, the striatum is the main target of dopamine neurons and projections to other areas are much weaker than projection to the striatum (Figure 1—figure supplement 3). Second, tracing studies of single dopamine neurons have shown that the striatum is the termination point of dopamine neurons, such that single dopamine neurons have a massive arborization in the striatum and that dopamine neurons which project to other brain areas do not pass through the striatum, with the notable exception of dopamine neurons that project to the cortex (Aransay et al., 2015, Cebrian and Prensa, 2010, Matsuda et al., 2009, Gauthier et al., 1999). Although the cortex shows much weaker labeling of dopamine axons than other areas (Figure 1—figure supplement 3), axons passing under VS potentially contributed to the signals in VS (Aransay et al., 2015). The axon tract from the midbrain to the forebrain runs along the ventral part of the brain, so any contamination of our signal was much more likely in VS than TS. Third, dopamine neurons that project to TS and VS are largely segregated and we did not observe axons passing through the opposite targets. Notably, dopamine neurons that target a particularly part of the striatum seldom pass through other parts of the striatum (Aransay et al., 2015, Cebrian and Prensa, 2010, Matsuda et al., 2009, Gauthier et al., 1999). Fourth, calcium transients in passing axons are much weaker than in axon terminals and boutons (Koester and Sakmann, 2000, Llano et 10, 1997). We added these points to our discussion (Discussion “Technical considerations”).

“Dopamine axons passing through and below the ventral striatum to the cortex (Aransay et al., 2014) may have contributed to the signals in VS dopamine, although calcium transients in passing axons are smaller than in axon terminals and boutons (Koester and Sakmann, 2000; Llano et al., 1997).”

*8) Regarding Figure 5: What happens if the authors sort the trials where there was licking versus no licking (in water trials) in terms of the response to the water delivery. No licking may indicate that the mouse did not recognize/detect/attend to the odor cue?*

We sorted our data based on anticipatory licking (rather than time) and plotted responses to new water-predicting odors and water delivery in VS and TS. We added a figure (Figure 5—figure supplement 2). Particularly this figure makes it clear that VS dopamine cue responses do not develop until after behavioral learning occurs. Additionally, it shows that TS cue responses were present before changes in behavior (Figure 5—figure supplement 2). This is in agreement with our proposal regarding “CS effectiveness” and “US effectiveness” (discussed below). Trial-to-trial correlation was also observed in a previous study of VTA dopamine neurons (Matsumoto et al., 2016). These data may support the relationship between responses and task engagement and/or learning, but we do not have direct evidence that they are related to attention and/or perception.

*9) It would be nice to see the anatomy: if the authors retrogradely label VTA DA neurons that project to VS versus TS, where do these TS neurons reside? What is the distribution? This would be helpful information in terms of joining this information with the work by Stephan Lammel on the arrangement of VTA DA neurons projecting to mPFC or NAc m shell or NAc l shell.*

We compared 8 subpopulations of dopamine neurons, including ones projecting to mPFC, and published this data in our previous paper (Menegas et al., 2015). VS-projecting DA neurons are mostly found in the VTA and TS-projecting DA neurons are mostly found in the lateral part of the SNC. This segregation along the medial-lateral axis could roughly coincide with the segregation of axons in the striatum along the anterior-posterior axis. We referenced this in the Discussion (“Relation to previous studies”).

“In our previous study, dopamine neurons that project to TS were mainly observed in the lateral SNc of mice (Menegas et al., 2015). A previous study proposed that putative dopamine neurons in the lateral SNc of monkeys encode ‘motivational salience’, which is the absolute value of positive or negative ‘value’ (Matsumoto and Hikosaka, 2009).”

*10) Did the authors control for the order of all of the manipulations? Were the order of manipulations randomized for both the trial types within A, B, C and the epochs (particularly in Figure 6)?*

Trial types in Figure 6 were randomized. Reward trials and nothing trials were used in all the sessions. In some sessions, either air puff trials or tone trials were added in addition to reward trials and nothing trials. Within a session, all trial types were randomized. Reward sessions, reward and air puff sessions, and reward and tone sessions were also randomized. We apologize for having poorly described our task parameters. We added an explanation of trial types and the order of sessions (Results (“Responses to rewarding and non-rewarding stimuli in VS and DS dopamine”) and Materials and methods (“Behavior”)). We also added a new section “Experimental Timeline” in the Materials and methods.

“After several weeks of this training, in some sessions, trials with odors predicting either a mild tone (55 dB) or an air puff were interleaved in addition to trials with water and trials with no outcome”.

“All trial types were randomized in all of the sessions”.

“After introduction of tone and air puff, reward sessions, reward and air puff sessions, and reward and tone sessions were randomized.”